



# Evaluation of the High Altitude Lidar Observatory Methane Retrievals During the Summer 2019 ACT-America Campaign

Rory A. Barton-Grimley[1], Amin R. Nehrir[1], Susan A. Kooi[2], James E. Collins[2], David B. Harper[1], Anthony Notari[1], Joseph Lee[2], Joshua P. DiGangi[1], Yonghoon Choi[2], Kenneth J. Davis[3]

[1] NASA Langley Research Center, Hampton, VA, USA
[2] Science Systems and Applications, Inc., Hampton, VA, USA
[3] Department of Meteorology and Atmospheric Science, and Earth and Environmental Systems Institute, The Pennsylvania State University, University Park, PA, USA

*Correspondence to*: R.A. Barton-Grimley (rory.a.barton-grimley@nasa.gov)

**Abstract.** The NASA Langley Research Center High Altitude Lidar Observatory (HALO) is a multi-function and modular lidar developed to address the observational needs of NASA's weather, climate, carbon cycle, and atmospheric composition focus areas. HALO measures atmospheric $H_2O$ mixing ratios, $CH_4$ mole fractions, and aerosol/cloud optical properties using the Differential Absorption Lidar (DIAL) and High Spectral Resolution Lidar (HSRL) techniques, respectively. In 2019 HALO participated in the NASA Atmospheric Carbon and Transport – America campaign on board the NASA C-130 to compliment

a suite of greenhouse gas in-situ sensors and provide, for the first time, simultaneous measurements of column $CH_4$ and aerosol/cloud profiles. HALO operated in 18 of 19 science flights where the DIAL and Integrated Path Differential Absorption lidar (IPDA) techniques at 1645 nm were used for column and multi-layer measurements of $CH_4$ mole fractions, the HSRL and backscatter techniques at 532 and 1064 nm, respectively, for retrievals of aerosol backscatter, extinction, depolarization, and mixing layer heights. In this paper we present HALO's measurement theory for the retrievals of column and multi-layer

$XCH_4$, retrieval accuracy and precision including methods for bias correction, and a comprehensive total column $XCH_4$ validation comparison to in-situ observations. Comparisons of HALO $XCH_4$ to in-situ derived $XCH_4$, collected during spiral ascents and descents, indicates mean difference of 2.54 ppb and standard deviation of the differences of 16.66 ppb when employing 15 s along track averaging (<3 km). A high correlation coefficient of R=0.9058 was observed for the 11 in-situ spiral comparisons. Column $XCH_4$ measured by HALO over regional scales covered by the ACT-America campaign are

compared against in-situ $CH_4$ measurements carried out within the planetary boundary layer (PBL) from both the C-130 and B200 aircraft. Favorable correlation between the in-situ point measurements within the PBL and the remote column measurements from HALO elucidates the sensitivity of a column integrating lidar to $CH_4$ variability within the PBL, where surface fluxes dominate the signal. Novel capabilities for $CH_4$ profiling in regions of clear air using the DIAL technique are presented and validated for the first time. Additionally, profiling of $CH_4$ is used to apportion the PBL absorption from the total

column and is compared to previously reported IPDA cloud slicing techniques that estimate PBL columns using strong echoes from fair weather cumulus. The analysis presented here points towards HALO's ability to retrieve accurate and precise $CH_4$ columns with the prospects for future multi-layer profiling in support of future suborbital campaigns.



# 1 Introduction

Atmospheric methane (CH₄) is a prominent greenhouse gas (GHG) with an increasingly important role in climate change due to rising emissions and their subsequent impact on radiative forcing. CH₄ has a global warming potential estimated to be 84 and 28 times greater than carbon dioxide ($CO_2$) over a 20 year and 100-year period, respectively (Myhre et al. 2013). Since pre-industrial times, CH₄ mole fractions have risen by 150% (Myhre et al. 2013) with the addition of anthropogenic sources identified as the cause of the rising abundance (Dean et al. 2018). CH₄ emissions can be apportioned between anthropogenic influences, such agriculture, waste management (Nisbet et al., 2016; Schaefer et al., 2016) and fossil fuel activities (Massakkers et al. 2016; Alvarez et al. 2018), and natural sources which are dominated by wetlands (Bousquet et al., 2006, 2011; Schaefer et al. 2016). Though the major sources of atmospheric CH₄ have been identified, uncertainty in emission rates (Ehhalt et al. 2001; Lu et al. 2022) detrimentally affects our understanding of the total CH₄ burden and its subsequent climate impact (Nisbet et al. 2014). Additionally, Lu et al. (2022) indicate that the time and spatial evolution of different emission sectors vary significantly across North America showing the need for continued atmospheric observations. The relative contributions and strengths of these highly varied sources require improved observations and increased spatial sampling to quantify these changing emissions.

The National Academies of Sciences, Engineering and Medicine (NASEM) 2017-2027 Decadal Survey for Earth Science and Applications from Space (NASEM 2018) called for further understanding of the sources and sinks of atmospheric CH₄, the processes that will affect their future abundances, and identified the need for improved measurement capabilities to advance the accuracy of climate models and inform policies that influence anthropogenic emissions. Jacob et al. (2016) discusses prominent methods by which atmospheric CH₄ can be measured from a satellite platform and the subsequent ability of these models to quantify emissions on regional and global scales is detailed. Passive measurements of column CH₄ from satellites (Frankenberg et al. 2011; Yokota et al. 2009; Hu et al. 2018) have been useful in many applications, such as large coverage inverse analyses (Wecht et al. 2014; Zhang et al. 2021) and regional emission analyses (Wecht et al. 2014; Zhang et al., 2020; Varon et al. 2020; Cusworth et al. 2021), the latter of which have been afforded by the high spatial resolutions of the most recently deployed sensors (Veefkind et al. 2012; Jervis et al. 2020). Despite the successes of these passive sensors, they are limited to daytime operation, have broad weighting functions that limit understanding of near surface fluxes, and suffer contamination from clouds, aerosols, and rapid changes in topography.

In-situ measurements have been used extensively for quantifying methane emissions. Useful accuracy and precision have been achieved when measuring emissions from cities (Cui et al., 2015; McKain et al., 2016; Heimburger et al., 2017; Plant et al., 2019; Lopez-Coto et al., 2020), and oil and gas production basins (Alvarez et al., 2018; Barkley et al., 2019) with an emerging ability to track emissions changes over time (Lyon et al., 2021; Lin et al., 2021). The in-situ measurement density available for this quality of emissions quantification, however, is limited at present to a small number of intensive study areas (Richardson et al., 2017; Verhulst et al., 2017; Karion et al., 2020). Global (Cooperative Global Atmospheric Data Integration Project, 2019) and continental-scale (Andrews et al., 2014) data collections exist, but their density limits the resolution and



accuracy of inverse flux estimates (Bousquet et al., 2006; Bruhwiler et al., 2014). Spatially dense observations from aircraft (Barkley et al., 2019b, 2021; Yu et al., 2021) exist and provide a robust data set that have great potential for improving quantitation of methane emissions, however their extent is limited to point altitude estimates.

Active sensing of atmospheric $CH_4$ can overcome many of the challenges that limit passive $CH_4$ and other GHG retrievals. Light detection and ranging (lidar) measurements of GHGs benefit from the direct generation of laser light to enable monitoring in all seasons, latitudes, during day and night, and allows for accurate measurements in the presences of clouds, aerosols, and topographic variability. Currently, no space instruments employing active techniques for GHG monitoring exist, however, development of the MERLIN (MEthane Remote sensing Lidar missioN) satellite, anticipated 2027 launch, (Ehret et al. 2017) will provide global measurements of $CH_4$ column-averaged dry-air mixing ratios ($XCH_4$) at 1.645 µm.

The differential absorption lidar (DIAL) method (Schotland et al. 1966; Schotland et al. 1974) is employed for the measurement of atmospheric $CH_4$ and other GHGs. At least two wavelengths of laser light are transmitted around a gas absorption line and differential attenuation through the atmosphere is experienced between the absorbing and non-absorbing wavelengths. The differential attenuation across a prescribed range bin can then be used to directly measure the GHG concentration, where the precision of the measurement is directly proportional to the size of the range bin. The integrated path 80   differential absorption (IPDA) technique, a variation of DIAL, provides high precision column-averaged dry-air mixing ratios of a GHG by utilizing strong echoes from clouds and the ground to measure the differential attenuation from the absorbing molecule of interest (Menzies et al. 2003, Ehret et al. 2008). IPDA offers high precision at the expense of profiling and has been demonstrated from airborne platforms as a highly precise and accurate method by which to measure total and partial column abundances of $CO_2$, $CH_4$, and other GHG (Riris et al. 2012, 2017; Dobler et al. 2013; Lin et al. 2015; Abshire et al. 85   2018; Refaat et al. 2020; Campbell et al. 2020). In preparation for the MERLIN mission, an airborne $CH_4$ IPDA demonstrator, CHARM-F (Amediek et al. 2017), has made progress towards demonstrating the expected measurement capabilities, targeted error budgets, spectroscopic requirements, and other research necessary to translate an IPDA lidar to spaceborne operation for global $CH_4$ measurements.

Recently, the NASA Langley Research Center (LaRC) developed a modular airborne DIAL/IPDA lidar to provide 90   multi-functional measurements of GHGs. The High Altitude Lidar Observatory (HALO) was developed as a more capable replacement for the NASA Lidar Atmospheric Sensing Experiment (LASE) $H_2O$ DIAL instrument (Browell et al. 1998) with improved operational flexibility and capability (Nehrir et al. 2018). HALO measures atmospheric $H_2O$ mixing ratios, $CH_4$ mixing ratios, and aerosol/cloud optical properties using the DIAL, IPDA, and high spectral resolution lidar (HSRL) (Hair et al. 2008) techniques, respectively. HALO was designed as an airborne simulator for future space-borne DIAL/IPDA missions 95   called for by the NASEM Decadal Survey (NASEM 2018) while also serving as a test bed for risk reduction of key technologies required to enable those future missions. To respond to a wide range of airborne science applications HALO can be rapidly reconfigured to provide $H_2O$ DIAL & HSRL, $CH_4$ DIAL/IPDA & HSRL, or $CH_4$ DIAL/IPDA & $H_2O$ DIAL measurements using three distinct modular laser transmitters and a single multi-channel and multi-wavelength receiver. First results from the $H_2O$ DIAL & HSRL configuration were discussed in Bedka et al. (2020) and Carroll et al. (2022). Here, we present results



from HALO's $CH_4$ DIAL/IPDA & HSRL configuration, which, to our knowledge, is the first ever demonstration of IPDA derived $XCH_4$ with simultaneous HSRL observations of aerosol optical properties. The coincident retrievals of $XCH_4$ and surrounding environmental contextual information (planetary boundary layer height (PBLH) and aerosol intensive/extensive properties) provides a comprehensive data generating capability which can be used for constraint of priors for inverse modeling of $CH_4$ fluxes to enable identification of sources, sinks, and inform large-scale transport models.

Novel to HALO is the ability to generate profiles of $CH_4$ DAOD, in addition to total column DAOD, using the DIAL technique. This retrieval was first demonstrated during the Long Island Sound Tropospheric Ozone Study (Judd et al. 2020). Traditionally this retrieval has been an inaccessible to $CH_4$ IPDA instruments due to weak molecular backscatter at 1645 nm (~1% of that at 532 nm) and a reduced ability to detect the weakly backscattered light due to poor detector performance at these spectral regions compared to readily available high gain components available at visible and NIR wavelengths. With

sufficient along track averaging HALO can generate relatively high signal-to-noise ratio (SNR) profiles of backscatter at 1645 nm, allowing access to preliminary range resolved retrievals. These retrievals have been evaluated for their feasibility and utility in apportioning the PBL region from the total column DAOD in addition to providing an alternate method to retrieve PBL mixing ratios in clear air regions where the cloud slicing technique (Ramanathan et al. 2015, Amediek et al. 2017) cannot be employed. Additionally, profiles of atmospheric backscatter at 1645 nm have been investigated as an alternative method

for total column IPDA bias correction without the need for in-situ spiral comparisons. The results present here are a preliminary assessment of retrieval performance and their application. A total quantitative assessment of the DIAL technique for $CH_4$ profiling will require high SNR not accessible to HALO currently due to detector limitations. Improved detector technology, such as advanced HgCdTe detectors (Sun et al. 2017), would enable routine profiling of lower tropospheric $CH_4$ for further evaluation and development of higher-level products. Despite lower detector performance, retrievals of IPDA offline

atmospheric backscatter have revealed detailed atmospheric structure that could be used for assessment of MLH in lieu of HSRL channels (currently retrieved from the 532 nm HSRL aerosol backscatter). Additionally, the backscatter could be calibrated (Fernald et al. 1984) to develop new intensive products, such as aerosol wavelength dependence between the 1645 nm and 1064 nm.

         This paper details the first results of HALO's $CH_4$ DIAL/IPDA & HSRL configuration from the 2019 NASA

Atmospheric Carbon and Transport - America (ACT-America) airborne campaign (Davis et al. 2021). It provides a brief overview of the measurement theory, instrument performance, examples of collocated $XCH_4$ and HSRL measurements, and introduces advanced methods to apportion $CH_4$ abundances within the planetary boundary layer (PBL) from the column with the DIAL technique. The paper is organized as follows: Section 2 provides a brief introduction to the HALO instrument and its measurement approaches. Section 3 gives an overview of the IPDA calibration process, methods to compare column

retrievals to in-situ validation measurements, bias correction, and performance analysis of $XCH_4$ precision and accuracy. Section 4 provides examples of retrievals at regional scales with comparison to PBL in situ measurements. Section 5 introduces advanced methods for range resolved profiling of $CH_4$ and direct PBL apportionment in clear air regions. Section 6 summarizes results and provides an outlook towards future impacts of HALO observations.



## 2 Instrument and Retrieval Description

### 2.1 Instrument Overview

HALO is a direction detection lidar which employs the DIAL/IPDA, HSRL, and standard backscatter techniques for measurements of GHG, clouds, and aerosols. The geometry for the combined DIAL/IPDA and HSRL measurement is shown in Figure 1. HALO is configured such that a single laser transmitter generates all of the requisite wavelengths for the $CH_4$ DIAL/IPDA (1645 nm), HSRL (532 nm) and backscatter (1064 and 1645 nm) measurements. The laser output is transmitted coaxially with a single collection telescope, from which the backscattered signals are collected and processed with a multi-wavelength receiver that houses conditioning optics, detectors, and control electronics. Specific details of the HALO instrument architecture will be presented in a future publication and the necessary details for retrieval are shown in Table 1.

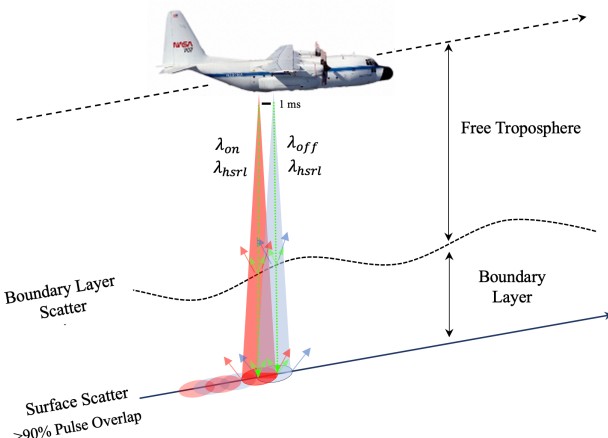

**Figure 1. HALO measurement geometry from the NASA C-130. Simultaneous acquisition of $CH_4$ DIAL/IPDA and HSRL data provides information about column $CH_4$ and aerosols/PBLH, respectively.**

HALO's $CH_4$ retrieval is carried out by interrogating the R6 line complex at 1645 nm. The 1 kHz pulse repetition frequency (PRF) laser light at 1645 nm is generated by a tunable optical parametric oscillator (OPO) (Nehrir et al. 2018; Fitzpatrick et al. 2019) which is pumped by a single frequency injection seeded Nd:YAG source at 1064 nm (Nehrir et al. 2018). Single frequency operation of the OPO is achieved by injection seeding two discrete continuous-wave distributed feedback (DFB) lasers that are spectrally stabilized to the online and offline spectral locations of the R6 line complex, 1645.5518 nm and 1645.3724 nm respectively. Injection seeding into the OPO cavity is done using fast electro-optical switches on a shot-to-shot basis, which results in a 500 Hz double pulse repetition frequency output from the OPO. The residual pump light, left over from the OPO conversion process, is frequency doubled to 532 nm after which the combined 1064 and 532 nm outputs are transmitted coaxially with the OPO output and used for the backscatter and HSRL retrievals. Injection seeding, combined with seed laser stabilization and pulsed laser cavity stabilization, ensures a high spectral purity of > 99.9% of the OPO and 1064 nm pump and allows high measurement accuracy and low bias. Monitoring the pulsed 1064 nm and 1645 nm outputs in real-time during flight operations, the peak frequency and width of each pulse, ensures optimal laser performance.

Figure 2 shows CH₄ absorption cross-sections at the R6 line complex calculated from the HITRAN 2016 database (Gordon et al., 2017) at two different pressure altitudes along with the transmitted DIAL/IPDA wavelengths (a Voigt line shape is assumed for all of the analysis presented herein). The online wavelength was selected in the trough of the line complex to provide uniform sensitivity to the lower free troposphere and reduce laser stability requirements, compared to operation at the peak of a single absorption line (Kiemle et al. 2011). The offline wavelength was determined by balancing the optimization of the CH₄ differential absorption optical depth (DAOD) and minimization of the H₂O DAOD.

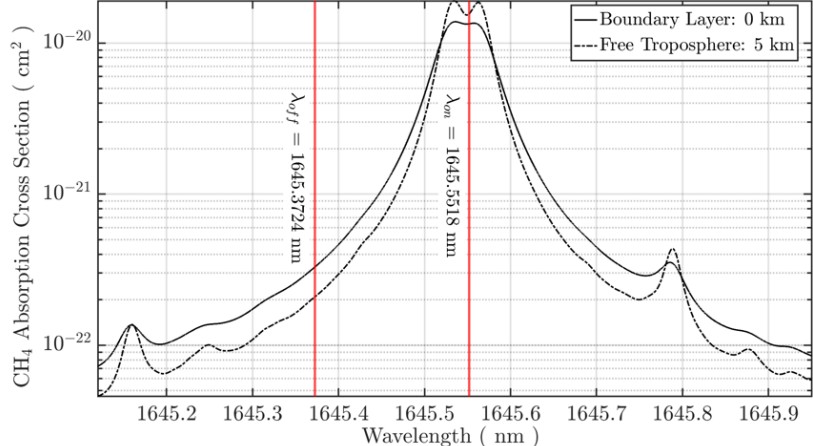

**Figure 2. Methane absorption cross sections calculated using a Voigt line shape for a standard atmosphere at 0 km and 5 km altitude. The online, 1645.5518 nm, and offline, 1645.3724 nm, wavelengths are shown in red.**

Unlike the DIAL technique, which does not require knowledge of the online and offline transmitted pulse energies, the IPDA technique requires accurate knowledge of these relative energy differences to normalize the backscattered signal from a scattering surface and calculate the CH₄ DAOD, which is then used to retrieve XCH₄. To capture the relative energy differences between transmitted pulses a laser energy monitor (LEM) subsystem samples a fraction of the transmitted beam, breaks speckle between laser shots (discussed further in Section 3.1.2), and detects the light with fiber coupled InGaAs avalanche photodiode (APD), equivalent to those in the receiver.

The received light is collected by a 0.4 m diameter all metal telescope, passed through a 0.65 nm interference filter to suppress unwanted solar background, and directed towards specific detection chains using dichroic splitters. The HALO CH₄ receiver chain employs three optical detection channels, one for boresight and two for science. The boresight channel directs a small amount of light to a quadrant PIN photodiode to maintain alignment between the transmit and receive paths and the remaining light is directed to the science channels. The linear dynamic range of the science channels is increased by splitting the light directed to the science channels such that one channel sees approximately 90% (high optical) and the second sees 10% (low optical) with separate detectors. The dynamic range is further increased by use of a dual buffered output from each detection chain with variable gain settings that cover a signal range exceeding 20 effective bits at the digitizer, or 60 dB. The large signal dynamic range allows for measurements over varying albedo, through tenuous clouds, and at varying standoff



distances from the scattering target without instrument reconfiguration or recalibration. The highest sensitivity channel, high-optical-high-electrical (HOHE), is used exclusively for atmospheric profiling at the CH₄ wavelengths, a unique feature of HALO. The remaining channels are utilized for the IPDA retrievals from cloud and surface returns; the high-optical-low-

electrical (HOLE) for high altitude operation and/or low albedo targets, the low-optical-high-electrical (LOHE) for mid-altitude operation, and the low-optical-low-electrical (LOLE) for low altitude operation and/or high albedo targets.

The native vertical resolutions for the DIAL/IPDA and HSRL channels are limited by the transmitted laser pulse widths (Table 1). The backscattered 1645 nm signals are digitized at a 120 MHz sample rate (1.25 m resolution in air) with a detection chain bandwidth of 3 MHz. To ensure that the transient response from the surface and clouds are accurately captured

the 1645 nm signals are retained at the 1.25 m vertical resolution for all IPDA calculations, serving to oversample the return pulse. The backscattered 1064 nm and 532 nm signals are digitized at 120 MHz sample rate with a 3 and 40 MHz detection chain bandwidth, respectively. To increase SNR and reduce the output file size the 532 nm signals are digitally filtered and both the 532 and 1064 nm data are decimated to 15 m vertical resolution. Figure 3 shows the ground return response at 1645 nm for a single 0.5 s profile at 1.25 m vertical resolution, where the HOHE profiling channel is fully saturated while the HOLE,

LOHE, and LOLE channels remain on scale for IPDA retrievals.

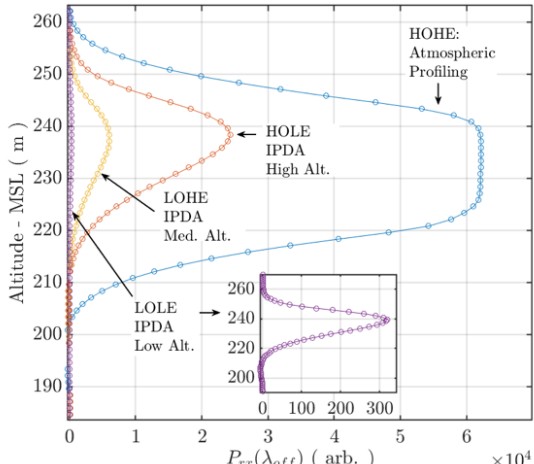

**Figure 3. Example of ground return impulse responses for the four IPDA receiver channels for a single profile. A single channel maintains high sensitivity for atmospheric backscatter. A combination of optical and electrical splits allows for optimization of the dynamic range to allow for sampling of the surface return backscatter over a wide range of aircraft altitudes and surface albedos.**

Because of the high PRF of HALO's pulsed laser, real-time onboard averaging is employed using field programmable gate arrays (FPGA) to further reduce the size of the recorded data file. The digitized signals are summed on the FPGA to a 2 Hz rate resulting in 500 accumulated shots for the 532/1064 nm channels and 250 shots at each wavelength for the 1645 nm channels. Although the data collection interval is 2 Hz, the high PRF transmitter ensures high pulse overlap exceeding ~94% overlap at high-altitudes (10 km) and ~87% overlap at mid-altitudes (5 km), considering a ~200 m/s aircraft speed. High pulse

overlap minimizes the effect of albedo variations between the online and offline IPDA samples and additional reduction of albedo variation noise to negligible levels is achieved by employing along track shot averaging (Amediek et al. 2009).



**Table 1. HALO parameters during ACT-America 2019**

| Parameter | |
|---|---|
| Laser Type | Fibertek: Nd:YAG pumped injection seeded OPO |
| Laser Wavelengths | 532 nm, 1064 nm, 1645 nm |
| Transmitted Laser Energy | 1.0 mJ, 2.5 mJ, 2.5 mJ |
| Laser PRF (532, 1064, 1645 nm) | 1 kHz, 1 kHz, 500 Hz double pulse |
| Laser Pulse Width | 5 ns (532 nm), 20 ns (1064 nm), 15 ns (1645 nm) FWHM |
| Spectral Purity | 532 nm: >99.98, 1645 nm: >99.96 |
| Laser Beam Divergence (1/e$^2$) | 0.8 mrad (532 nm), 0.8 mrad (1064 nm), 0.4 mrad (1645 nm) |
| DIAL/IPDA Wavelengths | 1645.5518 nm, 1645.3724 nm |
| HSRL/Backscatter Wavelengths | 532.2929 nm, 1064.5859 nm |
| DIAL/IPDA Vertical Sampling Rate | 120 MHz (1.25 m) |
| Effective Vertical Resolution | 15 m |
| Reporting Interval | 2 Hz (500 shot average 532/1064 nm 250 shot on/off average 1645 nm) |
| Collection Aperture | 0.4 m |
| Field of View | 1 mrad (532/1064 nm), 0.5 mrad (1645 nm) |

## 2.2 XCH$_4$ IPDA Measurement Technique

The range resolved 1645 nm backscattered laser light from the ground and clouds can be interpreted through the lidar equation for hard targets (W.B. Grant 1982). The received power at the digitizer from a target at a surface scattering elevation (SSE) is given by

$$P_{rx}(\lambda, R_{SSE}) = \frac{E_L(\lambda)}{t_{eff}} \cdot \left( \eta(\lambda)\beta(\lambda)\frac{A}{R_{SSE}^2} \right) \cdot e^{-2\left( \tau_{CH_4}(\lambda, R_{SSE}) + \Sigma\tau_g(\lambda, R_{SSE}) + \tau_m(\lambda, R_{sse}) + \tau_a(\lambda, R_{SSE}) \right)} + P_b(\lambda) , \qquad (1)$$

where the transmitted energy per laser pulse is $E_L$ (J) and the effective time domain response of the return signal is $t_{eff}$ (s). $\eta(\lambda)$ is a unitless wavelength dependent system constant that contains instrument efficiencies and all scalar values. $\beta(\lambda, R)$ is the target's reflection coefficient (sr$^{-1}$) and is equated as $\beta(\lambda) = \rho(\lambda)f(\lambda)$, where $\rho(\lambda)$ is the scatterer's reflectivity and $f(\lambda)$ is the bidirectional reflectance distribution function (sr$^{-1}$). The area of the telescope aperture is given by A (m$^2$), $R_{SSE}$ is the range to the scattering surface (m), and A/$R_{SSE}^2$ sets the solid angle of the receiver (assumes full geometric overlap of the transmitter and receiver). The exponential describes the two-way transmittance of laser light through the atmosphere and contains the optical depth terms $\tau_{CH_4}$, $\tau_g$, $\tau_m$, and $\tau_a$, which describe the extinction (absorption and scattering) due to CH$_4$ absorption, other absorbing gases, non-absorbing molecules, and aerosols. These terms can be understood through the Beer-Lambert law, where the optical depth due to CH$_4$ and the additional interfering gases over the measurement path is given by $\tau(\lambda, R_{SSE}) = \int_0^{R_{SSE}} \sigma(\lambda, r')n(r')dr'$, for a given absorption cross-section, $\sigma$ (cm$^2$), and gas number density, $n$ (cm$^{-3}$). The background molecular atmosphere and aerosol optical depth are defined by their respective extinction coefficients, $\alpha_m(\lambda)$ and $\alpha_a(\lambda)$ (m$^{-1}$). The solar background is given by $P_b(\lambda)$.



The digitized representation of the received power is proportional to the effective temporal response of the instrument and target. Under the assumption of Gaussian sub-components $t_{eff} = \sqrt{(t_L)^2 + (t_{det})^2 + (t_{tgt})^2}$ and is a geometrical sum of the FWHM temporal responses of the transmitted laser pulse width, $t_L$, the detection chain, $t_{det}$, and the scattering target, $t_{tgt}$. The detection chain response is composed of a total system bandwidth ($B_{sys}$), with contributions from the detector and post-detection amplifier, and can be approximated by $t_{det} \approx 1/(3B_{sys})$. The temporal response of the target, $t_{tgt}$, is

proportional to the terrain roughness and surface structure. From Eq. 1 the target's total power is estimated by integrating over $t_{eff}$ such that for each wavelength $P_{rx}(\lambda) = \int P_{rx}(\lambda, R')dR'$ is computed and is then used for IPDA retrievals.

To obtain the desired $CH_4$ measurement separate expressions of Eqn. 1 can be defined at the online and offline wavelengths and used to solve for the DAOD due to $CH_4$ as

$$\delta\tau_{CH_4} = \tau_{CH_4}(\lambda_{on}) - \tau_{CH_4}(\lambda_{off}) = \frac{1}{2}\ln\left(\frac{P_{rx}(\lambda_{off})}{P_{rx}(\lambda_{on})} \cdot \frac{E_L(\lambda_{on})}{E_L(\lambda_{off})}\right). \tag{2}$$

Equation 2 assumes that many of the variables from Eq. 1 are equivalent between the DIAL/IPDA wavelengths and cancel such that the DAOD is simply defined by the transmitted and received powers. A derivation of Eqn. 2 with no assumptions on the wavelength equivalence of terms can be found in Ehret et al. (2008).

The DAOD can be combined with atmospheric state parameters and a pressure weighting function to retrieve the column-weighted $CH_4$ dry-air mixing ratio as (Dufour and Bréon 2003; Ehret et al. 2017)

$$XCH_4 = \frac{\delta\tau_{CH_4} - (\delta\tau_{H_2O} + \delta\tau_{CO_2})}{\int_{p_a}^{p_{SSE}} w(p')dp'}, \tag{3}$$

where $\delta\tau_{CH_4}$ has corrections applied to account for the differential absorption of $H_2O$ and $CO_2$ (the two main interfering molecules) between the online and offline wavelengths, $\delta\tau_{H_2O}$ and $\delta\tau_{CO_2}$. To calculate $\delta\tau_{H_2O}$ and $\delta\tau_{CO_2}$, the relative humidity from reanalysis and a constant 400 ppm mixing ratio are used. In general, the contribution of DAOD due to $CO_2$ and $H_2O$ is negligible (on order of 0.0001 DAOD each), but still accounted for. The minimal impact from $\delta\tau_{H_2O}$ results from optimal

selection of the offline wavelength (Refaat et al. 2013).

Equation 3's weighting function is a description of the instrument's sensitivity to $CH_4$ absorption as a function of altitude and is explicitly dependent on the online and offline wavelength selection. At each pressure altitude the weighting function is defined as (Kiemle et al. 2011)

$$w(p) = \frac{\Delta\sigma_{CH_4}}{g(m_{dry} + m_{H_2O}q_{H_2O})}, \tag{4}$$

where $\Delta\sigma_{CH_4}$ is the $CH_4$ differential absorption cross-section (DCS), $g$ is the acceleration due to gravity, $m_{dry}$ is the average mass of a dry-air molecule, $m_{H_2O}$ is the mass of a water molecule, and $q_{H_2O}$ is the water vapor mixing ratio. Integration of Eq. 4 from the aircraft's altitude, $p_a$, to the SSE, $p_{SSE}$, gives the weighted average along the observed column. Figure 4 shows an example of a weighting function for HALO's spectroscopy, where near-uniform sensitivity can be seen across the lower troposphere and through the PBL.

HALO's retrievals of XCH4 are performed along the backscatter profile's slant path. The latitude and longitude of the ground spot for each measurement is realized by performing a geometric transformation from the transmitter to the SSE using the aircraft's global positioning system (GPS) and inertial measurement unit (IMU) data. This provides the surface pressure estimation at the SSE from the atmospheric state parameters. With an effective vertical range resolution of 15 m sampled at 1.25 m, the alignment of the calculated SSE with the GLOBE digital elevation model (DEM) (Hastings et al. 1998)

shows good agreement at 2 Hz and geolocation was deemed acceptable (an RMSE of 1.19 m over ocean is seen by HALO). Though HALO over samples the return pulse, Amediek et al. (2013) showed that it was possible to achieve <10 m ranging from a 150 m pulse and Ehret et al. (2008) showed that with $B_{sys}$ at the low value of 3 MHz would be sufficient to meet requirements for determination of the ground response.

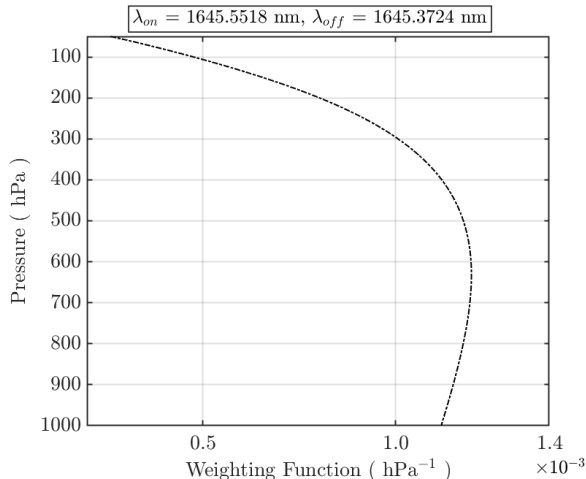

**Figure 4. Pressure weighting function used in the XCH4 retrieval for the HALO operating wavelengths in Table 1.**

The basic processing steps required to retrieve XCH4 are described by the flow diagram in Fig. 5. All calculations are performed from the basic quantities acquired during flight: transmitted power, received backscatter profiles, and the aircraft's IMU attitude and GPS timing information, the latter is used for geolocation of the SSE. The retrieval altitude grid is referenced to mean sea level (MSL) such that 0 m altitude is equivalent to the mean elevation of the sea surface (Altitude is used in lieu

of MSL for all figures). The time series of meteorological data inputs used to retrieve XCH4 from CH4 DAOD come from post-flight reanalysis. Vertically resolved pressure, temperature, and relative humidity curtains are generated along the GPS defined using NASA's Global Modeling and Assimilation Office's (GMAO) Modern-Era Retrospective analysis for Research and Applications, Version-2 (MERRA-2) (Gelaro et al. 2017). The analysis utilizes the 3-hour reanalysis product with all parameters converted to geometric height and vertically interpolated to HALO's resolution. To calculate the $CO_2$, $H_2O$, and

CH4 DCS the HITRAN2016 spectroscopic database is used (Gordon et al. 2017) with MERRA-2 pressure and temperature inputs. The DCS are then used for calculation of the weighting function, DAOD correction terms, and within in-situ derived XCH4 comparisons. Recent analyses for the MERLIN mission have shown that updates to the spectroscopy used in the XCH4 retrieval process (Delahaye et al. 2016, 2016; Vasilchenko et al. 2019) are required to overcome known biases in the line


parameters. This translates to retrieval bias and will be investigated for HALO retrievals in future analysis. The broad effects

of spectroscopy errors and the impact to retrievals are discussed in later sections.

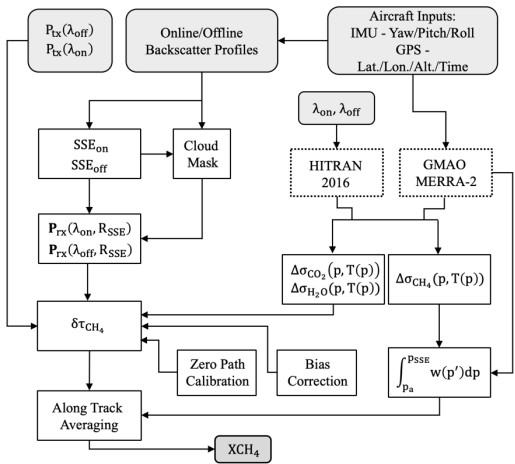

**Figure 5. Processing flow for the HALO IPDA XCH$_4$ retrieval.**

## 2.3 HSRL Measurement Technique

To provide additional information content and further context to the XCH$_4$ retrieval, HALO employs the HSRL

technique at 532 nm and traditional backscatter at 1064 nm. The methods and implemented architecture leverage developments

from prior NASA LaRC HSRL instruments (Hair et al. 2008). HALO utilizes an iodine vapor filter in the instrument's receiver

to separate backscatter contributions from the broadened molecular scatter, a few GHz in width, and the narrow Mie scatter

resulting from aerosols, which maintains nearly the same spectral distribution as the incident laser light, <100 MHz in width.

Utilizing the HSRL technique, aerosol extensive parameters – backscatter and extinction, and intensive parameters – aerosol

lidar ratio, aerosol depolarization ratio, spectral depolarization ratio, Angstrom backscatter coefficient, and aerosol typing can

be computed. Aerosol derived mixed layer heights are computed from the HSRL vertically resolved aerosol backscatter

product according to the methods discussed in Scarino et al. (2014). Explicit description of the HSRL techniques is provided

in Hair et al. (2008) and their use in HALO's H$_2$O configuration are further elaborated in Carroll et al. (2022), which mirrors

employment in CH$_4$ configuration.

**3 Airborne Measurements During ACT-America 2019**

HALO was integrated on the NASA C-130 aircraft in the summer of 2019 for the final ACT-America campaign

(Davis et al. 2021, Wei et al. 2021), where sorties were conducted out of Shreveport, LA, Lincoln, NB, and the NASA Wallops

Flight Facility, VA. During the campaign HALO's operation was limited to flight altitudes above the PBL to minimize

instrument exposure to the harsh temperature and vibration environments associated with increased temperature and turbulence

within the PBL. Comparison to in-situ instruments at regular intervals throughout the campaign provided a robust evaluation

of the accuracy and precision of HALO's CH$_4$ products.


### 3.1 Performance Analysis

For ACT-America HALO's DIAL/IPDA modality was operated in two configurations. The first utilized an attenuator in the transmit optical path to dynamically maintain signal linearity in the LOHE channel during flight. This has the effect of maintaining backscatter strength from all expected measurement altitudes and thus minimizing the probability of low-SNR retrievals on the LOHE channel. The second configuration transmitted the full laser power at all measurement altitudes and surface conditions; this configuration was exploratory and intended to exercise the full dynamic range of the receiver while providing a dataset by which to evaluate gain splicing of the different detection channels to account for changes in surface albedo and aircraft altitude.

As the IPDA technique relies on independent measurements of the transmitted pulse energy, accurate knowledge of the differential transmission between the transmit and receive path is required. Near field scattering effects on the differential transmission are ameliorated to the extent possible by placing the 1645 nm channel's field stop prior to the interference filter, which accounts for the largest source of differential transmission through the receiver (Nehrir et al. 2009). Measurement of the system's differential transmission is made by placing a scattering target in front of the transmit beam and collection aperture such that the receiver path is evenly illuminated without attenuation due to $CH_4$ absorption. We refer to this method as 'zero path' calibration. Many of these effects, and others not discussed here, were correctable with zero path calibration, repeatable over the duration of the mission, and have stayed stable since the initial instrument development. Additionally, we found that the zero path calibration term was independent of transmit power, allowing a single correction term to be applied throughout the entire campaign for each gain. The zero path calibrations were calculated for each receiver gain in pre- and post-campaign testing and removed from flight data to give the $CH_4$ DAOD as $\delta\tau_{CH_4}^{cal} = \delta\tau_{CH_4} - \delta\tau_{ZP}$. The average zero path calibration values were: 0.2971 (LOHE), 0.3128 (HOLE), and 0.2931 (LOLE). The superscript will be dropped for simplicity.

### 3.1.1 IPDA Optical Depth Bias Correction

Range dependent biases between the HALO DAOD and in-situ measurement-derived DAOD were observed during pre-campaign test flights. Similar biases of comparable magnitude and trend have also been observed in other airborne pulsed and continuous wave IPDA architectures (Campbell et al. 2020; Amediek et al. 2017; Fix et al. 2020). Studies examining the R6 line complex have shown that spectroscopic uncertainty can manifest itself as systematic bias in the retrieval of $CH_4$ from a remote sensor (Delayhe et al. 2016, 2016, 2019; Vasilchenko et al. 2016). Additional sources of error, such as laser spectral impurity, imprecise knowledge of transmitted wavelength, and other sources of systematic effect (Ismail et al. 1989) could potentially contribute to the observed range dependent bias, however, real-time characterization of the laser performance indicated that the laser transmitter was performing nominally. Sources of bias could also arise from intrinsic errors in the method of lidar to in-situ column comparisons, such as temporal phasing of the in-situ spiral relative to the lidar overpass (important when comparisons are in or near source regions) and misrepresentation of the total column by the in-situ



measurements due to the lack of observations at the surface. The latter spiral sampling issues were constraints of the mission and spectroscopic uncertainty are beyond the scope of this paper.

Test flights at the beginning and end of the campaign were utilized to compare HALO XCH$_4$ retrievals with in-situ derived XCH$_4$ and develop subsequent correction methods to remove the observed systematic bias. Stair-step descent maneuvers were employed followed by a descending spiral between each altitude leg for in-situ. Each stair step overflew the same ground track to generate multiple HALO DAOD estimates from fixed altitudes while observing the same air mass. A co-located Picarro spectrometer, calibrated to the WMO X2004A scale (DiGangi et al. 2021), on board the C-130 was utilized for in-situ

observations of the CH$_4$ mixing ratio. The lowest altitude of the spiral, ~300 m AGL, was filled in by extrapolating the last measurement to the ground to provide a complete profile from max flight altitude to the SSE. The in-situ CH$_4$ mixing ratio profile is converted to number density, combined with the HALO DCS (in-situ pressure and temperature profiles are utilized), and integrated from the respective altitude of each leg to the SSE. This generates a multi-point set of in-situ derived DAOD estimates from which the analogous HALO measurements can be directly compared to and any bias quantified. The potential

impact of near-surface variations in CH$_4$ were minimized by selecting locations that were distant from known point sources and by restricting maneuvers to the convective BL, such that vertical gradients close to the ground would be minimized.

To generate the bias correction terms a fractional difference between the mean in-situ derived DAOD, $\delta\bar{\tau}_{IS}$, and the mean HALO DAOD, $\delta\bar{\tau}_{CH_4}$, for each altitude leg is calculated as $y = (\delta\bar{\tau}_{CH_4} - \delta\bar{\tau}_{IS})/\delta\bar{\tau}_{CH_4}$. A single mean value for each DAOD time series over the entire altitude leg, with the average leg duration of < 5 min, is used to increase the accuracy of each DAOD

estimate. A relationship between $y$ and $\delta\bar{\tau}_{CH_4}$ for each altitude leg is then represented by a cubic polynomial model, $y = \beta_0 + \beta_1\delta\bar{\tau}_{CH_4} + \beta_2\delta\bar{\tau}_{CH_4}^2 + \beta_3\delta\bar{\tau}_{CH_4}^3$. A vector is then composed of the polynomial model for the entire maneuver, $\vec{y} = \boldsymbol{T}\vec{\beta}$, where $\vec{y}$ is the vector of fractional differences, $\boldsymbol{T}$ is the matrix of $\delta\bar{\tau}_{CH_4}$, and $\vec{\beta}$ is the vector of bias dependent correction coefficients. A least-squares regression solves for $\vec{\beta}$, which is then applied to correct the biased HALO DAOD as

$$\delta\tau'_{CH_4} = \delta\tau_{CH_4}[1 - \vec{\beta} \cdot \sum_{j=0}^{3} \delta\tau_{CH_4}^j]. \tag{5}$$

This method is similar to that developed within Campbell et al. (2020) for altitude bias correction of CO$_2$ IPDA estimates.

Figure 6 shows an example of a four-level stair step maneuver from the June 11$^{th}$ flight. HALO was operated in an 'attenuated' mode for this calibration maneuver, seen in the DAOD time series as a constant noise amplitude irrespective of flight altitude. Results from the 'unattenuated' mode of operation at the end of the campaign yielded comparable results. The native $\delta\tau_{CH_4}$ for all receiver gains is shown in Fig. 6b with $\delta\tau_{IS}$ overlaid. Fig. 6c shows the relationship between the native

HALO DAOD and the computed fractional difference with respect to the in-situ truth as a function of the fit parameters for each gain. The absolute fractional difference is approximately 2-2.5% for all altitudes, taken as the mean of all gains. Fig. 6d shows the resulting data with the altitude dependent correction applied, indicating that the fitting routine yields a zero-bias relative to $\delta\bar{\tau}_{IS}$. The 1-sigma error bars in Fig. 6d represent the DAOD uncertainty per gain channel due to shot noise, indicating that the fitting routine will yield lower uncertainty for optimized receiver gains.



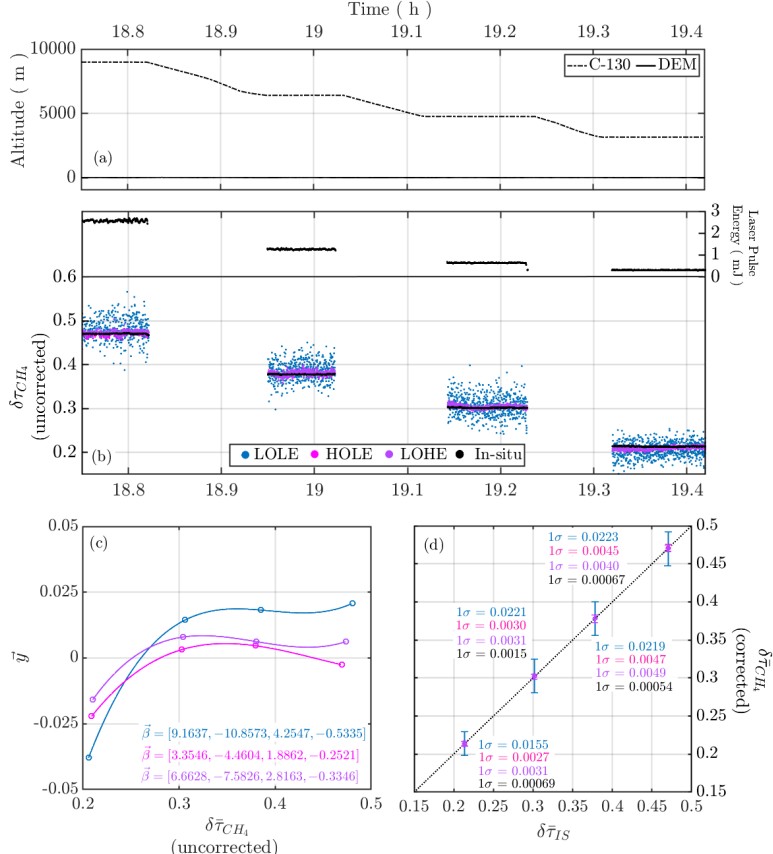


**Figure 6. Summary of the June 11$^{th}$ stair-step maneuver in Eastern VA and steps to calculate a DAOD correction. (a) Flight profile and DEM height. (b) 2 Hz HALO DAOD for all gains and the in-situ derived DAOD. The transmit pulse energies are shown indicating variable attenuation to maintain a constant surface signal amplitude. (c) Fractional differences between the mean HALO and in-situ derived DAODs for the different gains. (d) Bias corrected HALO DAOD compared to the in-situ derived values. Final 1-**
**sigma STD values for each point are shown with in-situ in black.**

The derived bias correction terms were applied uniformly across all data collected throughout the mission. Though each stair step maneuver generates only a few data points for fitting, favorable comparisons of bias-corrected HALO DAOD with in-situ observations throughout the campaign, as shown in subsequent sections (Figs. 10 & 11), demonstrates that the bias and correction was stable over the duration of the mission. Furthermore, this indicates the presented correction method offers

an interim solution to the observed biases while discrepancies in spectroscopy are investigated.

### 3.1.2 XCH₄ Retrieval

The HALO observables used to retrieve the column XCH₄ are shown in Fig. 7. The data span a 50 km along track flight segment for the low and high gain channels where the retrieval was optimized for the high gain. In each case the on and offline backscattered signals from the surface echo are digitized and summed on the FPGA. The integrated power from the

surface echo is estimated at each wavelength, $\boldsymbol{P_{rx}}(\lambda_{off})$ is shown in Fig. 7a & 7e. The peak of the georeferenced ground return



provides the SSE, shown in Fig. 7b & 7f in comparison to the DEM height. In this example, the SSE tracks the DEM closely, however, the optimized detection bandwidth and oversampling of the surface echo reveals the structure of the forest canopy. The integrated ground return is combined with the LEM measurement of pulse energies to calculate the DAOD according to Eqn. 3 and bias corrected with Eq. 5, shown in Fig. 7c & 7g. Finally, the DAOD and weighting function are combined according

to Eqn. 4, to retrieve XCH$_4$, shown in Fig. 7d & 7h. The aircraft's GPS coordinate system is used for all calculations and no additional steps are needed to align the 1645 nm backscatter to the DEM or MERRA-2 products.

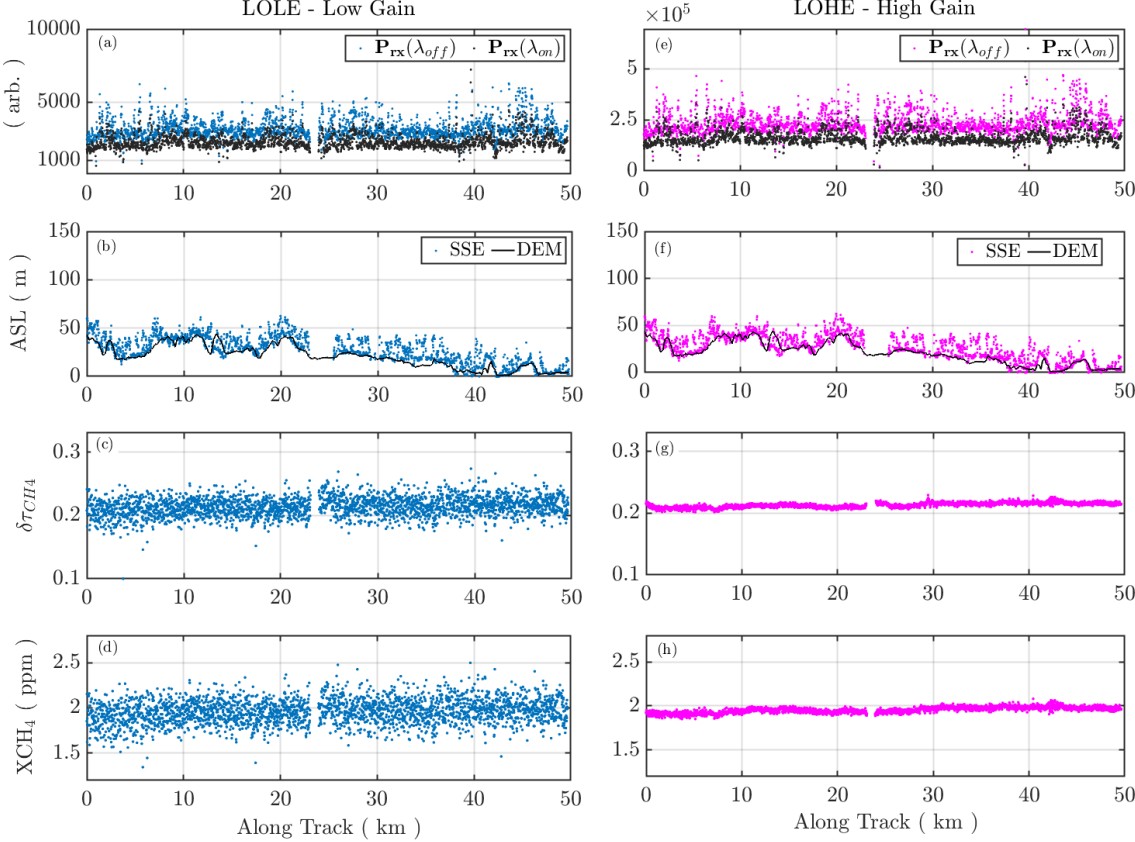

**Figure 7.** Example from the June 11$^{th}$ flight (3200 m AGL) of the macroscopic processing steps involved in retrieving the column XCH$_4$ for a 50 km along track segment at 2 Hz with the low gain and high gain channels shown on the left (a-d) and right (e-i), respectively. (a) & (e) show the integrated ground return signals. (b) & (f) show the calculated SSE and DEM. (c) & (g) show the calculated DAOD reported at 2 Hz interval. (d) & (h) show the final retrieved XCH$_4$ at 2 Hz.


The contrast in precision between gain channels in Fig. 7 is indicative of the SNR dependency of the XCH$_4$ retrieval and offers the ability to optimize the retrievals over a large dynamic range. The 30-40 km along track portion of the high gain column XCH$_4$ from Fig. 7h is further examined in Fig. 8, where histograms of the 2 Hz retrieved data from the optimized high

gain channel are shown against a 15 s averaging window for comparison. The 1-sigma standard deviation (STD) along this section gives 19.825 and 8.257 ppb, respectively and indicate a high precision at short averaging scales.





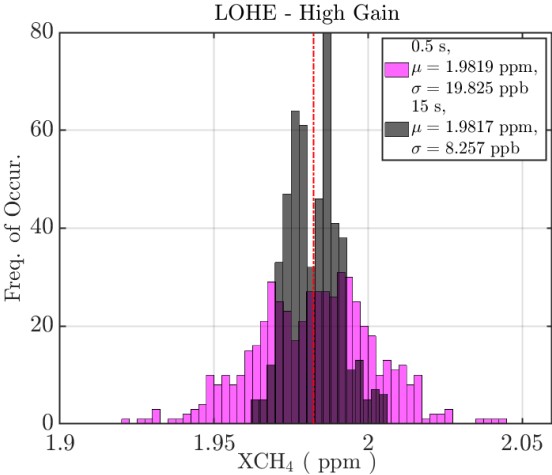

**Figure 8. Histograms of the along track XCH₄ retrievals, 30-40 km from Fig. 8h. Raw 2 Hz (0.5 s) data is shown in comparison to a 15 s average for the optimized high gain signal. In each case the mean value and 1-sigma standard deviation are shown.**


It was found that acceptable precision for all gain channels, ≤10 ppb, was reliably achieved with 15 s averaging windows. This was applied to all retrievals discussed here and was used to overcome noticeable decreases in precision experienced periodically throughout the campaign. To examine the retrieval precision the 1-sigma standard deviation with different averaging times is computed, often described as the Allan Deviation. Figure 9 shows an example of the noise statistics

calculated from several flights across the central, southern, and eastern United States which exhibited varying surface structure, albedo, and flight altitude. Retrievals using a DAOD calculated from the non-optimized low gain channel show a ~1% std (< 20 ppb) with <10-15 s of averaging and ~0.5% (<10 ppb) with 10-20 s of averaging. Retrievals made using optimized regions from the high gain channel show a ~1% std with ~1-5 s of averaging and ~0.5% with 5-10 s of averaging. Further averaging increases precision for applications that require high sensitivity, such as identifying weak emissions in thawing boreal regions.

Although high precision can be achieved with relatively short averaging times, and different gains are employed to allow operational flexibility, the performance observed during ACT-America fell short of prior flights on the Langley B200 aircraft. The increased statistical noise observed could result from the harsh operating conditions on the C-130, resulting from slightly degraded laser frequency stability due to the high vibration environment.

Another contributing factor to the higher statistical uncertainty observed during ACT-America could result from

speckle introduced by the long coherence length of the pulsed laser transmitter. HALO minimizes speckle within the receiver in two ways, first through the receiver by employing large collection apertures and secondly by employing along track shot averaging, the latter of which will inherently break speckle cell correlation at the collection aperture on a shot-by-shot basis. On the transmitter, the correlation of speckle cells must be broken between subsequent laser shots to measure the online/offline energy ratio of the transmitted laser pulses accurately, which is one of the main challenges of IPDA (Fix et al. 2018).






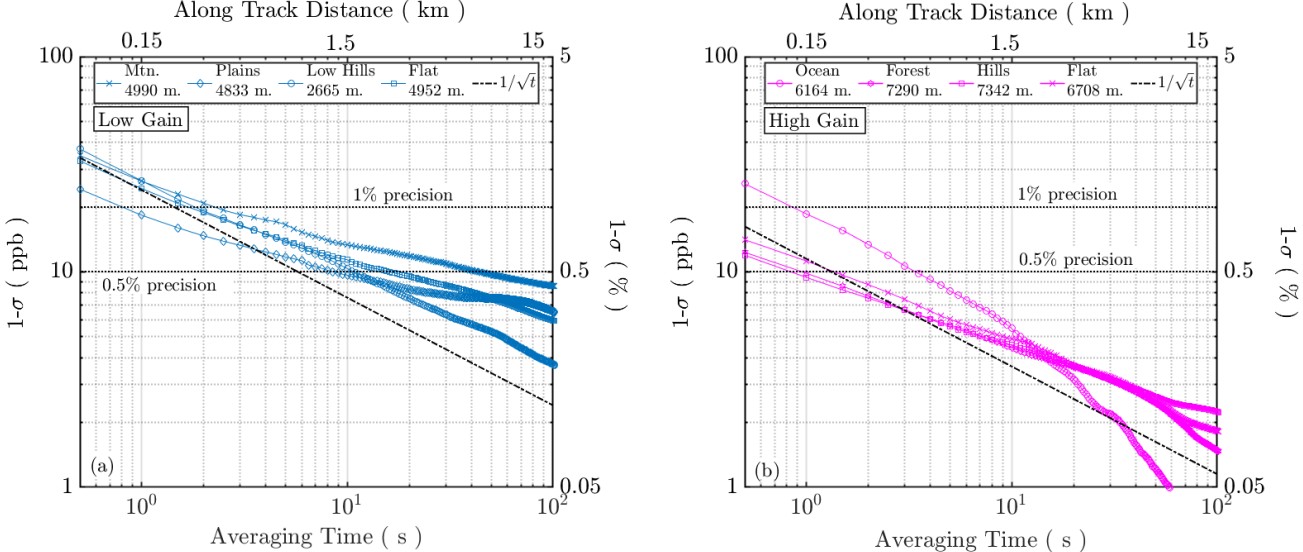

**Figure 9. XCH₄ noise statistics from the low gain, (a), and high gain, (b), for different terrains. The XCH₄ measurement precision at the native 2 Hz interval is ~ 10-50 ppb depending on terrain conditions and channel optimization. With a 15 s averaging window, ~2 km along track, measurements approach a <10 ppb precision, ~0.5% assuming a 2000 ppb background. The along track distance assumes a 150 m/s ground speed.**

HALO's LEM employs a similar energy measurement method as reported in Fix et al. (2018). First, two integrating spheres are used to attenuate the sampled pulse to acceptable levels. A multi-mode optical fiber further attenuates the light circulating within the second integrating sphere and is used to transport the sampled pulse to the LEM detector. Diffusers are placed at the input aperture of the first and second integrating spheres and are used to break the correlation of speckle cells introduced by the rough surface of the integrating spheres themselves. The relatively small diameter of the collection fiber (105 μm) and slow oscillating frequency (180 Hz) of the speckle reducing diffusers, compared to the 1 kHz PRF of the pulsed laser, results in residual speckle cell correlation between the online and offline over several pulses. Zero path calibration indicates that the speckle limited noise floor of the DAOD measurement is limited to ~0.005 over a half second average (250 shots/wavelength), where additional averaging provides further reduction. A recent MERLIN study (Cassé et al. 2019) showed that the impact of speckle on transmit energy measurements scales with SNR and that the expected random noise due to speckle for MERLIN approached <=5 ppb (or ~0.25% for 2000 ppb) with <10 s of averaging. These values are in line with HALO's findings and indicate the potential for speckle to dominate measurement noise if not accommodated for. Future investigations to further reduce speckle in HALO's LEM measurements are under investigation.





## 3.2 In-situ Validation

Vertical profiles of GHGs ($CO_2$ and $CH_4$ amongst others) and meteorological variables were periodically sampled in-situ on each aircraft and offered a unique validation opportunity. An overpass of the in-situ profile location prior to, or after, the C-130 spiral, descending or ascending, allowed for direct comparison of the lidar derived $XCH_4$ to in-situ derived $XCH_4$.

An example of a spiral maneuver from the July 20th flight from ~5.2-0.3 km AGL and a ~12.5 km diameter overpass of the spiral is shown in Fig. 10. A 3D representation of the inbound and outbound flight line, overpass, and in-situ $CH_4$ measurements are shown in Fig. 10a. The in-situ profile of $CH_4$ mixing ratio is interpolated to HALO's vertical grid, shown in Fig. 10b in black, and is then used to derive an in-situ $XCH_4$ retrieval from each flight altitude, shown in Fig. 10b in magenta. Comparing the mixing ratio profile to the in-situ derived XCH4 in Fig. 10b facilitates an understanding of the differences

between a point measurement at a given altitude and the equivalent column weighted estimate from that altitude. In Fig. 10b, the highest in-situ derived $XCH_4$ retrieval (~5.2 km AGL) provides the comparison value to HALO's estimate. The mean HALO $XCH_4$ retrieval from the overpass is also shown in Fig. 10b at 1.9086 ppm with an 9.46 ppb STD and compares to the in-situ derived $XCH_4$ estimate of 1.9001 ppm with an STD of +/- <1 ppb. This gives a mean difference of 8.5 ppb, or 0.447%, indicating that HALO has good agreement with the in-situ measurement.

Each C-130 ascent or descent spiral profile that met requirements for lidar comparison (e.g., wings level, stabilized pulsed laser, low cloud extent) was used to evaluate HALO's $XCH_4$ retrievals. After screening, 11 of 23 spiral profiles (9 descent, 2 ascent) were used in comparison to HALO $XCH_4$ from the coincident overpasses. In some cases, spiral ascents were performed after long duration boundary layer legs, resulting in an inability for the OPO to stabilize prior to the post ascent overpass, others had inadequate overpasses for HALO sampling. Some comparisons were carried out from a low flight altitude

which can limit lidar measurement precision (i.e., precision increases proportionally with DAOD). For each comparison a manual selection of the gain channel was used to optimize SNR. Figure 11 shows the correlation of the in-situ $XCH_4$ from the spiral profiles to HALO's $XCH_4$ from the coincident overpasses. Each point is colored by the HALO DAOD and has a designation for spiral direction (ascent vs. descent). A correlation of R=0.9058 was calculated for all comparisons and we define the bias across all comparisons as the mean difference between HALO and the in-situ derived estimate, giving 2.54

ppb, and a 1-sigma standard deviation of the differences of 16.66 ppb. It should be noted that the comparison to in-situ during stair step maneuvers used for bias correction are not included within this comparison and no additional calibrations were applied to the data collected throughout the campaign.



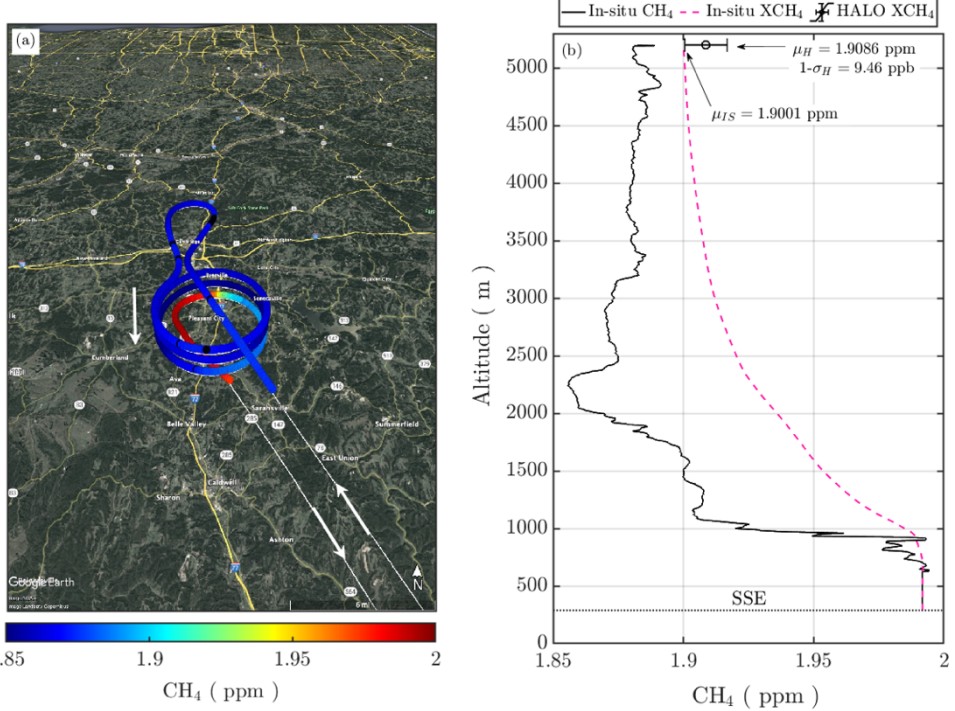

**Figure 10. (a) C-130 flight track and in-situ CH$_4$ profile from the July 20$^{th}$ flight (© Google Maps). (b) in-situ profile of CH$_4$ mixing ratio as measured during a descent spiral from approximately 5 km to the ground height (SSE), in black. Overlaid is the in-situ derived XCH$_4$ in magenta using HALO's weighting function. The in-situ derived XCH$_4$ at flight altitude was $\mu_{IS} = 1.9001$ ppm, shown as the top point of the magenta curve, and the black error bars shows HALO's overpass mean value with $\mu_H = 1.9086$ ppm and 1-$\sigma_H = 9.46$ ppb.**

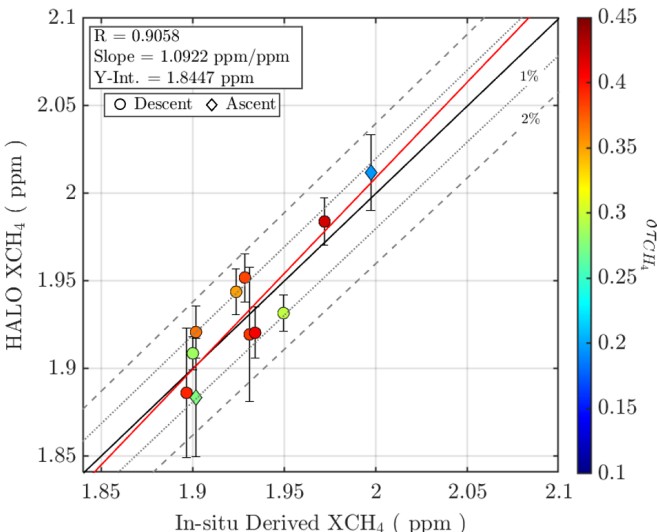

**Figure 11. Comparison of the in-situ derived XCH$_4$ to HALO XCH$_4$ for 11 spirals, color coded by the HALO one-way DAOD. A correlation between in-situ and HALO gives R=0.9058, with the fit shown as a red dashed line against the black one-to-one line. The 1% and 2% error bounds are shown. The 1-sigma error bars are shown for the HALO XCH$_4$ retrievals.**





The locations of the spiral maneuvers analyzed in Fig. 11 were planned to be distant from strong local sources

whenever possible. However, we expect that it is possible that a comparison could have unexpected enhancements below the

minimum aircraft spiral altitude which are not represented within the in-situ profile. Though unlikely, this could account for

some of the differences seen between the two instruments. To understand such a scenario, and the subsequent impact of on an

in-situ derived column estimate, the profile in Fig. 10b is further examined. For a uniform 50 ppb enhancement added to the

range bins from the lowest spiral altitude to the SSE (approximately 400 m), the in-situ derived column estimate changes by

only ~4.5 ppb from 1.9004 ppm to ~1.905 ppm, a 0.25% increase. Interpreted through Fig. 11's results the aggregate mean

difference between HALO and in-situ decreases by < 1 ppb, indicating that this effect is likely not a major driver of the spread

in random error. This does, however, emphasize the challenge in validation and evaluation methods for a column integrating

lidar, where enhancements not captured in-situ, but seen by the lidar, would translate to few ppb changes over the total column

and would be comparable with the total allowable systematic error, the example here accounting for one half.

## 500 4 Regional Scale Observations

ACT-America's regional sampling strategy and coordinated flights between the C-130 and B200 aircraft provided a

unique opportunity to evaluate HALO's observations to in-situ data over large regional scales. Near spatially coincident flight

lines for C-130 and the B200 aircraft are shown in Fig. 12 from the July 20th flight. The spatial and temporal coordination

between the two aircraft during this flight provided an ideal opportunity to assess the sensitivity of the HALO column $XCH_4$

measurements to variability within the PBL where surface fluxes dominate signals. Due to differing flight speeds, altitudes,

and B200 refueling, the alignment of the two aircraft in time is offset until the latter portion of the flight, with the C-130

lagging the B200 by ~2 hr. at the start to the C-130 forward of the B200 by ~0.5 hr. at the end. HALO's $XCH_4$ and coincident

HSRL aerosol backscatter are shown in Fig. 12a & 12b from the C-130 and in-situ PBL $CH_4$ from the B200 in Fig. 12c. The

associated HALO IPDA path length with the temporal separation of the two aircraft overlaid is shown in Fig. 12d. Screening

of the B200 in-situ measurements to the PBL utilized a combination of HALO's MLH and examination of the B200's in-situ

water vapor mixing ratio for transitions to the moist PBL, ≥14 g/kg. Figure 12a & 12c show good spatial agreement for the

enhancements and magnitudes between HALO's column $XCH_4$ retrievals and the PBL in-situ observations. Several regional

enhancements (e.g., urban, agricultural, oil/gas) were observed by both instruments and these spatially covarying signals

provide qualitative indication that HALO's column $XCH_4$ has sensitivity to PBL $CH_4$ abundances.






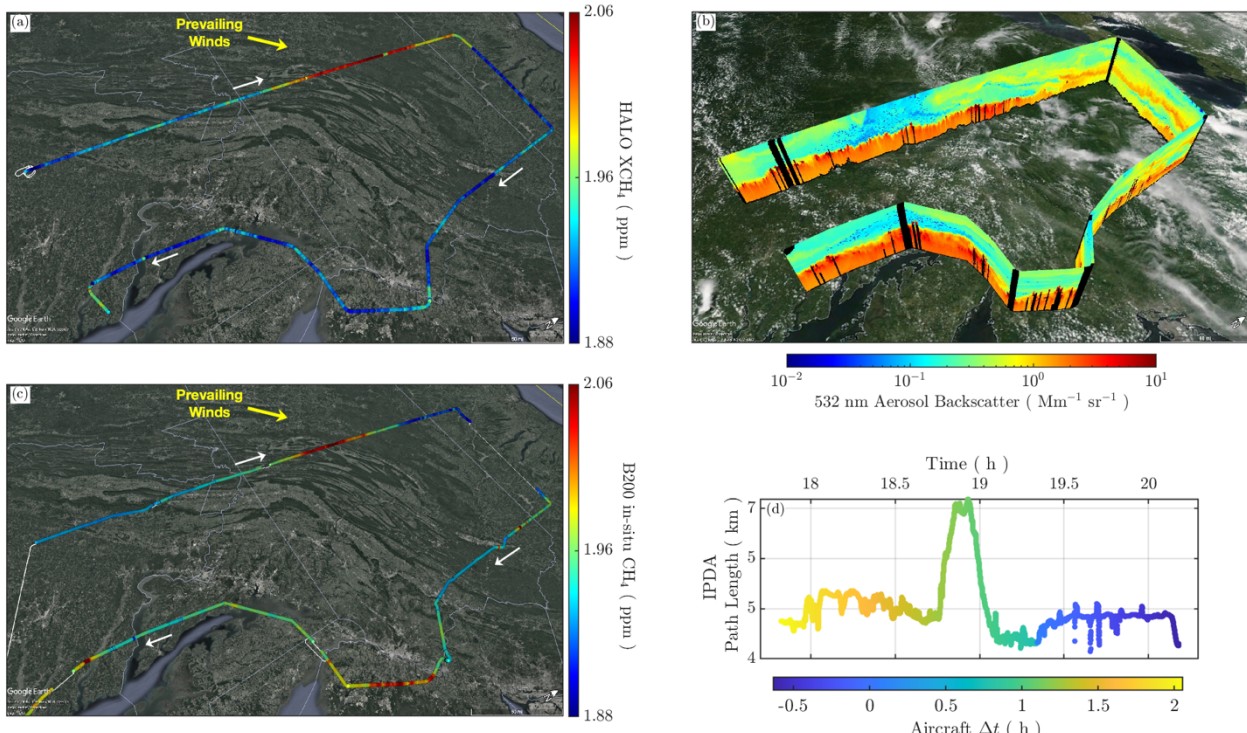

**Figure 12. Comparisons between HALO's column XCH₄ from the C-130 and in-situ PBL CH₄ from the B200 during the second leg of the July 20ᵗʰ flight. (a) Cloud cleared HALO XCH₄ retrievals (© Google Maps). (b) HSRL derived aerosol backscatter at 532 nm with an overlay of the TERRA MODIS corrected reflectance to indicate cloud extent (© Google Maps). (c) In-situ sampled CH₄ restricted to the PBL (the B200 aircraft landed for refueling, ~18.5-19.25) (© Google Maps). (d) IPDA path length with the C-130 and B200 temporal separation.**

Of particular interest is the S-N transect of the PA region, where a significant enhancement is observed by both instruments. This broad enhancement is likely explained by emissions from the regional natural gas and coal production facilities (Barkley et al. 2019). The transect is expanded in Fig. 13, where the time series of HALO XCH₄, in-situ B200 PBL CH₄, and C-130 FT CH₄ are shown in Fig 13a. At the lower latitudes of the transect HALO and the PBL in-situ agree to within 25-50 ppb of each other (~1-2% difference), indicating that little to no enhancement is present within the lower troposphere and that the absolute magnitude of the column measurements correlate well with point measurements. A steady regional enhancement, maximizing at ~150 ppb above background, is seen by HALO and in-situ from southern PA, 40° N. to northern PA, 42° N. Given the HALO weighting function it is expected that HALO's measurement of the enhancement would be expected to be muted compared to the PBL in-situ observations (like Fig. 10b). At the latter portion of the transect (north of ~41.2° N) the in-situ enhancement subsides to background levels while HALO still measures a ~75-100 ppb enhancement. These differences could arise if the FT air has elevated CH₄ originating from a different source than the more local emissions captured by the PBL observations. This hypothesis is supported by the appearance of an elevated aerosol layer in Figure 12b that appears at approximately 40° N, the point where the HALO XCH₄ appears to increase with distance along the flight more rapidly than the in-situ mole fractions (Figure 13a). Closer examination of this layer in Fig. 13b shows that an inflow of air



lofting aerosols into the FT is present, with the B200 in-situ wind direction within the PBL indicating a south-westerly flow in the PA enhancement region. This elevated aerosol layer potentially originates from a PBL source far upwind of the flight line, and thus may include elevated $CH_4$ mole fractions. This could explain the divergence between the PBL and column $CH_4$ measurements, particularly at the northern end of the flight track. These results show the sensitivity of $XCH_4$ measurements

to advected enhancements, similar to the conclusions of Feng et al. (2019a,b) concerning $XCO_2$ observations. These comparisons demonstrate the value of HSRL in detecting these advected layers, the need for atmospheric transport models to interpret these data more fully, and the potential value of $CH_4$ profiling. Additional analysis with model comparisons, such as those conducted in (Bell et al. 2020) for $XCO_2$, are required to definitively attribute the total column enhancement and will be the subject of future investigation.

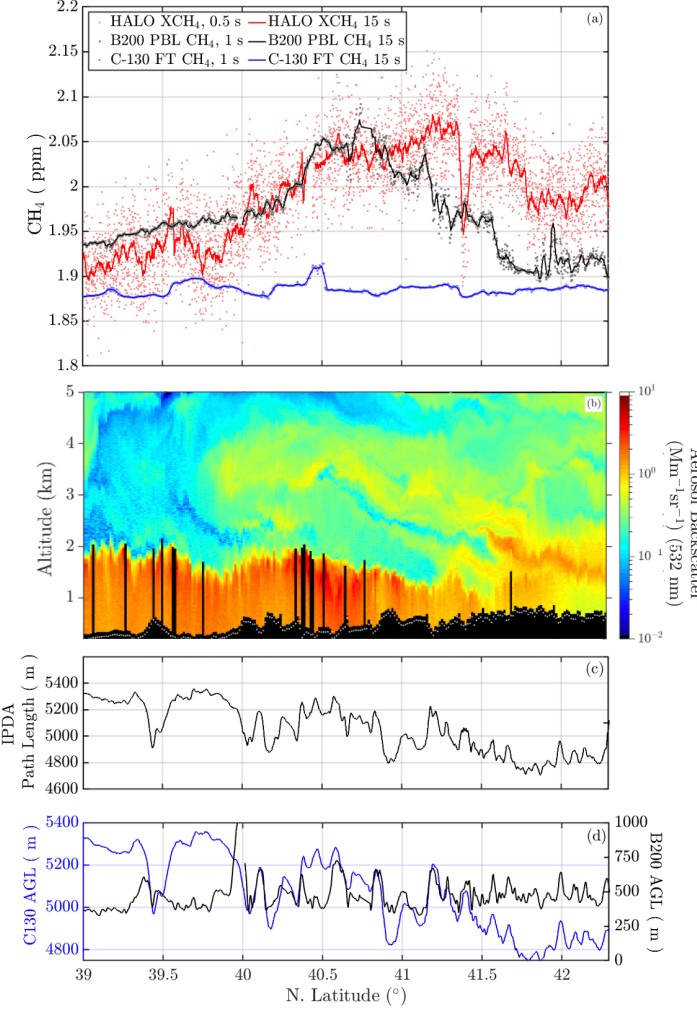


**Figure 13. South-north transect, 39-42.3°N, of Pennsylvania from Fig. 12. (a) Mixing ratio measured by each instrument, HALO column $XCH_4$ in red, the B200 PBL in-situ $CH_4$ in black, and C-130 in-situ FT $CH_4$ in blue. (b) cloud cleared (black vertical lines) HSRL aerosol backscatter at 532 nm during the transect and DEM height, white dots. (c) IPDA path length. (d) Altitude above ground level (AGL) time series of each aircraft.**





To further assess the ability of a lidar column measurement to observe variability from near surface emissions the HALO $XCH_4$ was correlated to the PBL in-situ observations from the B200 for the full flight, as in Fig. 12, as well as the PA S-N transect, as in Fig. 13. The comparisons were limited to 0.1° (~8 km) radial search between the HALO total column and B200 PBL data. Given a planned flight line overlap this filter ensures the nearest latitude/longitude of each aircraft is used for

comparison. The spatially filtered data yields a correlation of R=0.3507 for the full flight and R=0.4003 for the S-N transect. Apart from the PA S-N transect, HALO underestimates the B200 observations on a whole but still captures the variability seen by the B200, however, this is expected as column averages exhibit influence from lower background values out of the PBL. Despite the relatively variable time separation and mismatch of sampling volumes between the two data sets, the correlation coefficients indicate mild correlation is present and further demonstrates the ability of column integrating measurements to

observe PBL variability.

         A second example of a comprehensive data set for comparison was collected on the June 27[th] flight in the southern portion of the Mississippi River Valley. The comparison was divided into two comparison regions, $XCH_4$ vs C-130 PBL $CH_4$ and $XCH_4$ vs B200 & C-130 PBL $CH_4$. The first comprised the western leg where the C-130 flew at altitude (~6.5 km) to collect $XCH_4$ then in the reverse direction within the PBL (<1 km above ground level) to sample the same background region

in-situ. The second comprised the southern legs where the C-130 flew in the FT to make in-situ and $XCH_4$ measurements in coordination with the B200 sampling $CH_4$ in the PBL. The C-130 subsequently flew within the PBL for in-situ sampling on a northern return to provide an indication of how the PBL enhancement changed spatially. Figure 14a shows all regions of coincident HALO $XCH_4$ and PBL in-situ $CH_4$, from each aircraft for both regions. Here, HALO is at altitude within the FT and the C-130 and B200 legs are within the PBL. Multiple regions show covariance between the remotely sensed column and

the in-situ PBL observations. Figure 14b shows the lower central region where the enhancement comparisons take place. Here all instruments register enhancements, emphasized by their alignment in the cross-hatched flight line areas. Given the location of these flight lines it is likely that these plumes are indicative of wetland emissions. To provide further context a curtain of the HSRL aerosol backscatter with the overlaid PBLH in red is shown in Fig. 14c. Signals attenuated beneath opaque clouds are masked out in black and provide insight into the atmospheric state during the sampling time. In the earlier portion of the

day background comparison region portion of the flight exhibits a shallower PBL (pre-Noon local standard time), whereas in the latter portion of the day where the enhancement comparisons occur a deeper PBL has developed, and significant aerosol lofting has occurred. In-situ measurement of the PBL wind direction during the second comparison indicates winds flowing from the NW as measured during the C-130 PBL leg at the end of the comparison window.





**Figure 14. Overlapping flight lines between the C-130 and B200 aircraft in the southern Mississippi River Valley region from the June 27th flight. (a) Combined flight lines for low altitude PBL in-situ observations from C-130 and B200 and HALO XCH₄ from the C-130 at high altitude (© Google Maps). (b) Strong regional correlation can be seen between all three instruments, HALO at high altitude and the C-130 and B200 PBL observations at low altitude (© Google Maps). Prevailing winds measured in-situ within the PBL indicate a NW flow during the observation period. (c) HSRL aerosol backscatter for the high-altitude C-130 legs with the PBLH overlaid.**



The previously described correlation approach was applied to the two comparison regions from Fig. 14a & 14b. The stacked legs out and back on the background comparison region exhibit close to zero correlation at R = 0.0792, indicating that
column measurements made in background conditions void of large emissions do not correlate with weaker surface fluxes and that elevated signal captured by HALO was not sampled in-situ on the Eastern return leg within the PBL. The cross-hatched flight lines sampled at high and low altitudes by the remote and in-situ instruments, respectively, encompassed by the eastern part of the comparison region from Fig. 14b demonstrates a higher degree of correlation at R = 0.7218 between the B200 PBL measurement and HALO and R=0.4290 between the C-130 PBL measurement and HALO. The combined correlation analysis
of HALO to both PBL in-situ instruments in the enhancement region exhibits a correlation of R=0.6075. The PBL wind direction measured in-situ by the C-130 and B200 indicate a complicated wind scene with a general NW flow in the enhancement region, and low wind speeds of 5-10 knots. Examining Fig. 14b, the measurements made within enhancement region the in-situ $CH_4$ indicates a delineation between background and the enhancement, and despite a difference in absolute magnitude the spatially defined enhancements captured by all instruments provide further indication that column derived $XCH_4$
measurements can be used as an indicator for PBL enhancement.

## 5 Advanced $CH_4$ Products – Atmospheric Profiling

The DIAL technique uses ratios of atmospheric signals to derive a relative DAOD and the number density within a prescribed range interval. Using atmospheric signals directly, DIAL is self-calibrating and overcomes many of the challenges associated with IPDA to generate a column measurement (zero-path calibration, bias correction, and reference energy
measurement). Benefits of higher precision are also afforded with DIAL as the retrieval is non-linearly proportional to the range bin size (Nehrir et al. 2017, Carroll et al. 2022) such that the large vertical averages required to increase the per bin number of photons will also increase the $CH_4$ DAOD precision (SNR values in excess of 500 are required for highly precise DIAL/IPDA retrievals). Although absolute knowledge of the total DAOD is not needed for a typical DIAL retrieval, here we have chosen to normalize the backscattered signals throughout the profile to near aircraft signals to compare the atmospheric
derived cumulative DAOD to the IPDA column DAOD.

Coincident measurements of the range corrected offline backscattered signal and the HSRL 532 nm aerosol backscatter for the duration of the July 20th flight are shown in Fig. 15, where the offline backscatter was spatial averaged to 15 m vertical resolution and 10 s along track to match the HSRL retrieval resolution. The two data curtains qualitatively demonstrate the ability of the DIAL/IPDA channels to capture key atmospheric features needed to enhance IPDA column
measurements with profiling capabilities. Figure 15c shows the vertical profiles of HSRL aerosol and offline backscatter collected over the spiral overpass region analyzed in Fig. 10. The profiles indicate that the offline SNR is sufficient for a range resolved retrieval, however retrieval quality and effectiveness is limited by the online wavelength's optical depth. This is further examined within Fig. 16.

To perform a range resolved DIAL retrieval and estimate profiles of DAOD from the 1645 nm online and offline
backscattered signals a modified version of Eqn. 2 is used as





$$\delta\tau_{CH_4}^{DIAL}(R) = \frac{1}{2}\ln\left(\frac{P_{rx}(\lambda_{off},\ R)/P_{rx}(\lambda_{off},\ R_{norm})}{P_{rx}(\lambda_{on},\ R)/P_{rx}(\lambda_{on},\ R_{norm})}\right).$$ (6)

Unlike the IPDA derived DAOD, the range resolved calculation utilizes backscatter profiles which have been normalized by atmospheric signal from the nearfield of the aircraft, $R_{norm}$. The normalization signal's altitude is chosen such that full geometric overlap has been achieved while also ensuring that appreciable CH4 DAOD has not accumulated in the

bins. This provides a comparable method to estimating the cumulative DAOD over the lidar profile for comparison to traditional IPDA estimates and without an ancillary LEM module for characterizing the difference in online and offline pulse energies. In practice, the best placement of $R_{norm}$ could still yield non-negligible amounts of CH4 DAOD between the aircraft and the normalization point, <0.01 for the comparisons during ACT-America. For a robust comparison to IPDA this additional optical depth must be estimated and included within the cumulative estimate per range bin. When present, this is estimated by

calculating the DAOD difference between the nearest signal to the aircraft and the normalization bin as $\delta\tau_{CH_4}^{DIAL}(R_{norm}) - \delta\tau_{CH_4}^{DIAL}(R_a)$, which is then added to each bin of the DAOD profile. The benefit of a range resolved DAOD profile calculated with Eq. 6 is that no bias correction is applied and the energy differences between pulses are measured within the atmospheric profiles.

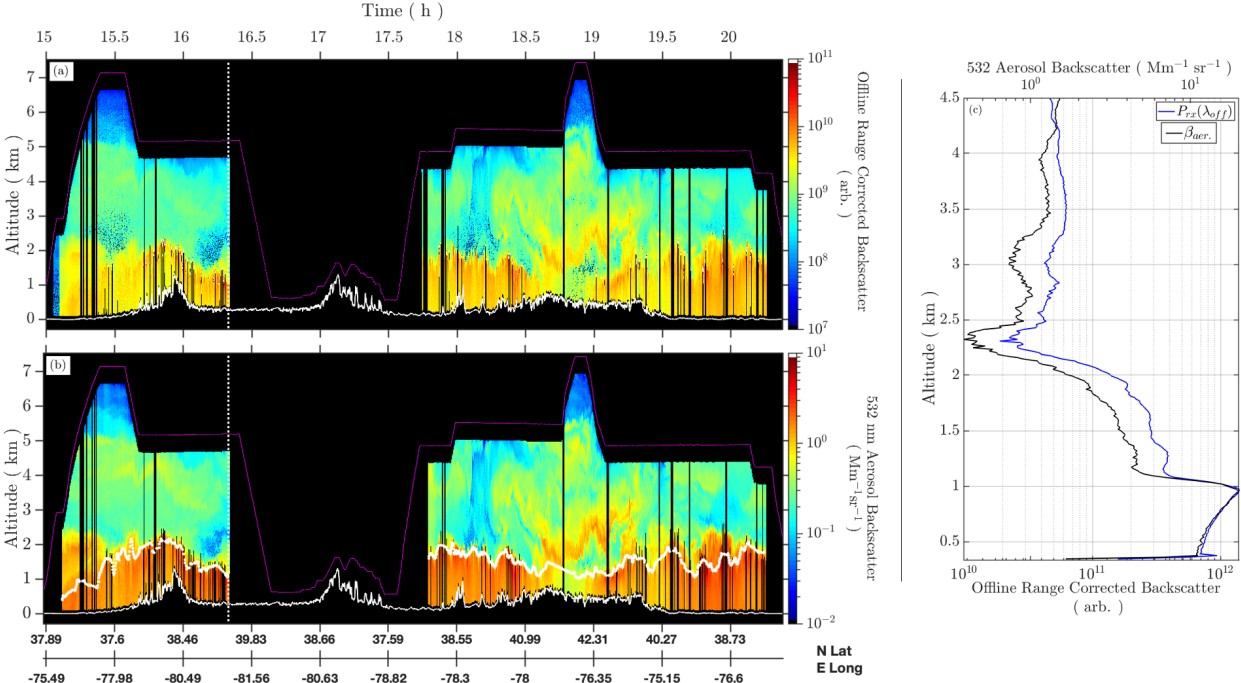

**Figure 15. Examination of 1645 nm offline backscatter from July 20th. (a) Range corrected offline backscatter profiles (15 m; 10 s). (b) HSRL aerosol backscatter at 532 nm (15 m; 10 s) with MLH in white. The flight track, magenta, and DEM height, white, are shown in each panel. Each curtain was cloud cleared with the HSRL cloud top height, black striations. (c) Profiles of offline backscatter and the HSRL aerosol backscatter from the vertical white lines shown in panels a-b (15 m; 12.5 km).**





To investigate CH₄ profiling capabilities the spiral overpass presented in Figs. 10 & 15 was further examined, allowing simultaneous comparison of the in-situ derived DAOD, HALO IPDA derived DAOD, and HALO DIAL derived DAOD. Figure 16a & 16b shows subsections of the range corrected offline and online backscatter centered about the spiral location, where differential absorption between the DIAL/IPDA wavelengths can clearly be seen within the PBL backscatter. Figure 16c shows $\delta\tau_{CH_4}^{DIAL}(R)$ calculated with Eq. 6 for the duration of the overpass. To increase SNR and enable retrieval with

Eq. 6 the input backscatter profiles were averaged to 350 m vertical and 15 s along track (2 km). Increasing DAOD can be seen from the FT into the PBL with an average value in the lowest retrieved bin approaching ~0.275 (one-way DAOD). Additional features can be seen within the DAOD curtain that correlate with the aerosol field, such as the clear air feature at ~2.5 km at the latter section of the overpass. This feature appears to be a manifestation of low-SNR in this region, resulting from low aerosol backscatter and larger standoff distance to the aircraft, and not the result of decreased CH₄ optical depth.

The online and offline backscatter signals were further aggregated over the entire overpass window to 350 m by 12.5 km to increase precision and a single range resolved retrieval was made. Figure 17a shows the input backscatter profiles and the DAOD profile is shown in Fig. 17b. Here the near linear trend in the lower tropospheric DAOD is fully observed and is the result of the uniform weighting of absorption due to pressure broadening of the line complex in the lower atmosphere. The inset in Fig. 17b shows the DIAL and IPDA derived column estimates along with the in-situ derived DAOD from the overpass'

spiral. Due to the required vertical averaging for the DIAL retrieval the last atmospheric bin above the SSE is unresolved, setting the accumulated DAOD in the lowest retrieved atmospheric bin at 0.2723. To provide a comparable estimate to the IPDA derived value at the SSE, a linear regression was performed on the DIAL calculated profile and extrapolated to the SSE, shown in Fig. 17b, giving an estimate of $\delta\tau_{CH_4}^{DIAL}(R_{SSE}) = 0.2943$. This contrasts the IPDA and in-situ estimates of $\delta\tau_{CH_4} = 0.2837$ and $\delta\tau_{IS} = 0.2829$ at $R_{SSE}$ and indicates that the DIAL derived DAOD overestimates the total column estimates,

IPDA and in-situ, by 3.66 & 3.95 %, respectively. The magnitude of the differences between the two independent measurements are on order of the differences between the un-bias-corrected IPDA DAOD and the in-situ derived DAOD shown in Fig. 6 and provides further insight into the uncertainties associated with the CH₄ line parameters/spectroscopy used in the derivation of in-situ derived XCH₄ and within HALO XCH₄ retrievals.





**Figure 16. 1645 nm range corrected backscatter (15 m; 10s) from the July 20th in-situ overpass subsection of Fig. 15. (a) Offline,**
**1645.3724 nm. (b) Online, 1645.5518 nm. The white bars in each panel indicate the overpass of the spiral location, see Fig 10. Each curtain was cloud cleared and the DEM is overlaid. (c) shows the calculated range resolved DAOD at 350 m vertical and 15 s of along track averaging within the overpass region defined in (a) and (b). The lowest retrieval bin occurs one 350 m range cell above the DEM due to the large vertical retrieval window.**





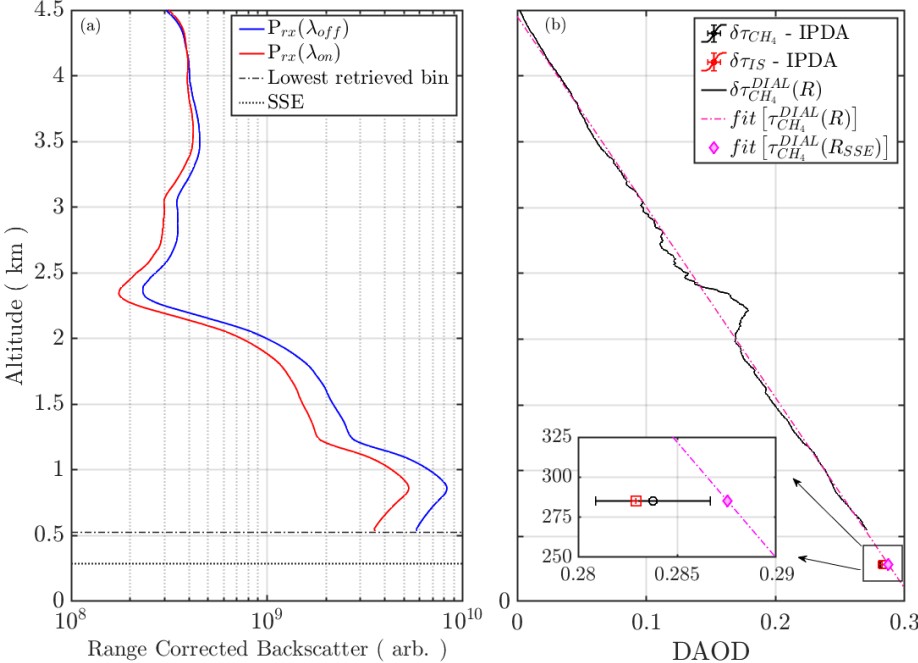

**Figure 17. (a) The 1645 nm range corrected online and offline over the overpass region averaged to 350 m vertical by 12.5 km along track resolution. (b) The range resolved DAOD as the black line, with fitted DAOD from the range resolved profile shown by the magenta dashed line. The magenta 'diamond' emphasizes the fitted value at the SSE, the IPDA derived DAOD as the black circle with 1-sigma error bars, and in-situ derived DAOD as the red box.**

## 5.1 Planetary Boundary Layer Apportionment

Traditional methods for apportioning the PBL mole fraction from IPDA column measurements have relied on the "cloud slicing" technique (Ramanathan et al. 2015, Amediek et al. 2017). This method requires that fair weather cumulus and stratocumulus clouds cap the PBL and that IPDA columns measured to surface and cloud top can be subtracted and used to infer abundances of GHGs within the PBL, $\delta\tau_{PBL}^{IPDA} = \delta\tau_{ground}^{IPDA} - \delta\tau_{cloud}^{IPDA}$. Though this method has shown utility in retrieving near surface mole fractions, its usability diminishes in regions and conditions void of clouds. Figure 18a shows the time series of IPDA DAOD surrounding the overpass in Fig. 16, where fair weather cumulus clouds at PBL top prior to the overpass provide lower DAOD estimates and changes in SSE translate directly to changes in DAOD. Histograms for the entire window are shown in Fig. 18b, binned in DAOD increments of 0.001, where the distributions of DAOD at cloud top and ground are clearly delineated and enable an estimate of $\delta\tau_{PBL}^{IPDA}$ using the cloud slicing method. From the histograms mean values from each DAOD distribution were estimated as $\delta\tau_{ground}^{IPDA} = 0.2848$ and $\delta\tau_{cloud}^{IPDA} = 0.2164$, and the subsequent PBL DAOD of $\delta\tau_{PBL}^{IPDA} = 0.0683$. Though this shows the ease at which $\delta\tau_{PBL}$ can be computed directly from the DAOD time series when clouds are present, the methods are restricted to the presence of clouds and findings are extrapolated to the clear air overpass. Given that the overpass region is mostly cloud free, the prior clouds provide the information required for comparison between $\delta\tau_{PBL}$ estimates derived from cloud slicing and the clear air overpass region can be computed through a DIAL. Combined,





both methods bring about the potential for complementing measurements in variable atmospheric states and allow a contiguous measurement throughout cloudy and cloud-free regions.

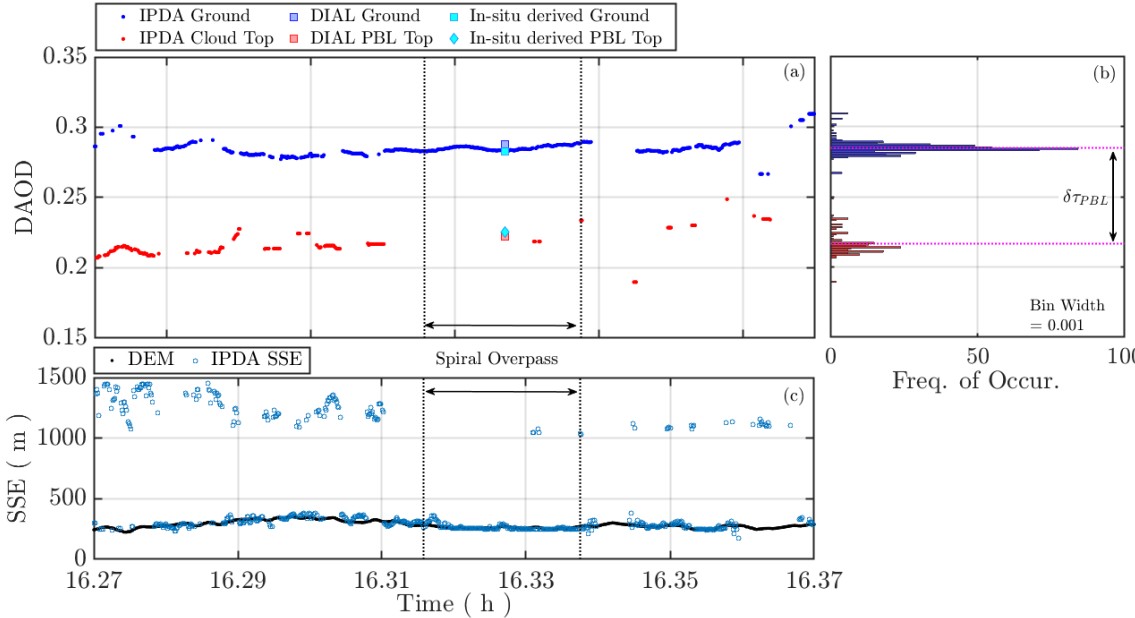

**Figure 18. Summary of PBL DAOD derivation for the overpass shown in Fig. 16. (a) Shows the column and cloud top DAOD surrounding the overpass region with DIAL derived and in-situ DAOD estimates of overlaid. (b) Average DAOD from cloud top and to the SSE, where histograms indicate the peak estimates from which the PBL abundances can be derived using cloud slicing, giving $\delta\tau_{PBL}^{IPDA} = 0.0683$ for the duration of the window. (c) Compares the IPDA SSE ground and cloud top height to the DEM.**

For the clear air region, $\delta\tau_{CH_4}^{DIAL}(R)$ can be used to estimate the DAOD PBL top which can then be can be subtracted from the total column DAOD to give the relative PBL contribution as $\delta\tau_{PBL}^{DIAL} = \delta\tau_{CH_4}^{DIAL}(R_{SSE}) - \delta\tau_{CH_4}^{DIAL}(R_{MLH})$. Given HALO's measurement modalities the HSRL derived MLH can be used to discern the PBL top and the IPDA SSE can for the ground elevation, indicated in Fig 19. From the DAOD fit the DAOD at each altitude can be extracted to give a DAOD estimate for the PBL column as $\delta\tau_{PBL}^{DIAL} = 0.0557$. Comparing to the in-situ derived DAOD for the portion of the column, $\delta\tau_{PBL}^{IS} = $

0.0561 was estimated from the spiral profile when using HALO's HSRL MLH and IPDA SSE as integration bounds. The relative components for the PBL column each computation are shown within Fig. 18 in contrast to the cloud slicing estimate. Utilizing the in-situ temperature and pressure profiles from the spiral a subset of the HALO weighting function for the PBL was used to derive a PBL column mixing ratio of 1.9629 ppm, from $\delta\tau_{PBL}^{DIAL}$, and 1.9775 ppm, for $\delta\tau_{PBL}^{IS}$. This gives a difference of ~0.741% and indicates that the HALO DIAL method has the potential to provide clear air estimates of PBL XCH₄.

Further examination of $\delta\tau_{PBL}^{DIAL}$ and $\tau_{PBL}^{IS}$ indicates that they differ from the estimate derived using cloud slicing, the latter of which appears to provide an absorption overestimate when extended to the clear air region. This is likely due to differing mixing ratios between the air masses such that extrapolation is not valid, or spectroscopy induced error resulting from



application of the correction described in section 3.1.1 to the base IPDA retrievals used within the cloud slicing computation.
Figure 18 indicates that absorption estimates to PBL top and over the total column are consistent for all three methods, despite

exhibiting minor differences. When estimating PBL specific absorption however, small PBL DAOD uncertainties, even on the

order of 0.001-0.002, can translate to several percent uncertainty in the derived geophysical observable XCH$_4$.

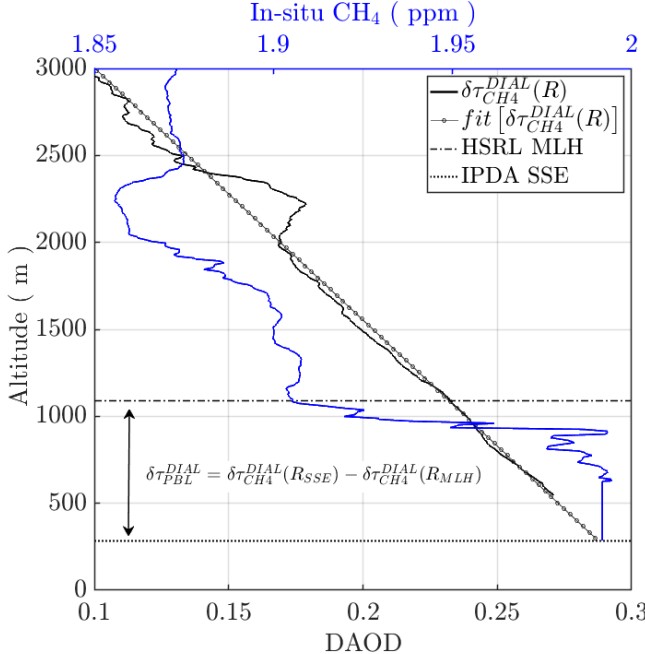

**Figure 19. Range resolved DAOD at 350 m vertical by 12.5 km along track resolutions, black, with fitted estimate overlaid, grey, with extrapolation to the SSE. The HSRL MLH and IPDA SSE are shown for the overpass region and allow an estimate of $\delta\tau_{PBL}^{DIAL}$**
**from the fitted profile. The in-situ CH$_4$ profile from the subsequent spiral profile, blue, is shown for comparison.**

Utilization of this method for future PBL focused studies requires further development to document uncertainties

from the DIAL retrieval. As shown here, the DIAL retrieval suffers from lower SNR compared to the IPDA retrievals, which

benefit from the strong surface returns. For the cases where high DIAL SNR can be achieved with moderate along track

averages (requiring increased PBL backscatter or significantly improved detection methods), this retrieval could provide new

insights on PBL fluxes in clear air regimes. To complement the DIAL retrieval and extend measurements down to the surface

without the need for linear fitting and extrapolation, a hybrid-IPDA (HIPDA) method has been devised which utilizes the

atmospheric signals at PBL top and the strong surface return to directly apportion the PBL DAOD from the column (e.g.,

filling in the 350 m above the SSE). The HIPDA method is similar to that employed in HALO's WV DIAL retrievals (Carroll

et al. 2022), where the DAOD due to WV between the lowest retrieval bin and the SSE is estimated and used to extend the

DIAL derived mixing ratio through the entire PBL. HIPDA is currently being adapted to the CH$_4$ retrieval, however, validation

of the technique has not been performed and will be the subject of a future publication.



## 6 Conclusion

The HALO $CH_4$ DIAL/IPDA measurements were quantitatively evaluated for the first time during the 2019 ACT-
America campaign. Data was collected from the NASA C-130 aircraft during 18 of the 19 flights and 2 engineering flights.
These flights were the first detailed validation efforts of a combined $CH_4$ DIAL/IPDA and HSRL, demonstrating a unique
ability to contextualize $CH_4$ column measurements with additional information afforded by the HSRL and backscatter profiles.
Data collected during this mission provided a unique opportunity for validation and assessment of instrument stability and
retrieval accuracy and precision. Additionally, the data provided insight into future investigations, such as optimization of
spectroscopic line parameters which currently serve as the largest source of uncertainty in the HALO $XCH_4$ retrieval.

Analysis across the duration of the campaign found that the single point calibration of HALO's $CH_4$ channels coupled
with the overall stability of the HALO instrument, provided repeatable and reliable measurements of $XCH_4$ over a wide range
of atmospheric and surface conditions aboard an environmentally challenging aircraft. Data collected over varying terrain were
used to compute noise statistics for the high and low gain channels and showed that a precision of 0.5% was achievable for
averaging intervals of <15 s in the low gain channel and <10 s in the high gain channel, allowing for operation at different
aircraft altitudes and over different surface albedos. Comparisons of HALO to in-situ derived column estimates were carried
out throughout the campaign, where in-situ profiles were generated during spiral ascents or descents under the overpass region
and provided validation of HALO's $XCH_4$ measurements. An overall correlation of R=0.9058 with a bias across all
comparisons, the mean difference between HALO and the in-situ derived estimate, of 2.54 ppb and a 1-sigma standard
deviation of the differences of 16.66 ppb across all 11 comparisons was observed. Given HALO operated in vastly different
research modes to optimize for emerging atmospheric profiling $CH_4$ retrievals an improvement in reducing the required along
track averaging to achieve consistent <1% precision is expected in future flights. This can be achieved by optimizing the
transmit energy (or the receiver optical splits between different gain channels) to better utilize the high optical signals for the
IPDA measurement. Lastly, several comparisons of lidar derived $XCH_4$ and in-situ measurements of $CH_4$ within the PBL were
made at regional scales and showed high degrees of covariance. These demonstrated the ability of a column integrating lidar
to observe $CH_4$ variability within the PBL where $CH_4$ fluxes dominate signals.

An altitude dependent bias of < 2.5 % (average) was identified in HALO's DAOD when compared to in-situ. These
biases were removed by correcting the lidar measurements to in-situ truth through a stair step maneuver carried out in
background conditions assumed void of known enhancements. A single set of corrections was applied to each channel for the
entirety of the campaign. The resulting bias-corrected data showed excellent agreement with in-situ spiral profiles for the
campaign duration, demonstrating the instrument stability and validating the correction method employed. The bias source has
been investigated and all indicators point towards an uncertainty in the spectroscopic line parameters derived from HITRAN
2016. Our findings here agree with findings published in preparation for the MERLIN mission (Delahaye et al. 2016;
Vasilchenko et al. 2019) as well as those found by the CHARM-F IPDA lidar instrument (Fix et al. 2020). Future work will
incorporate updated spectroscopy into the $XCH_4$ retrievals, and a bias reduction/removal is anticipated.



During the 2019 ACT-America flights HALO demonstrated for the first time in a scientific setting range resolved measurements of $CH_4$ DAOD employing the DIAL technique. The DIAL technique can overcome the primary challenge associated with IPDA, namely the requirement of accurate knowledge of the transmitted energy ratio and receiver/transmit path differential transmission ratio, which serve as the two largest sources of uncertainty in an IPDA lidar. Longer horizontal averages than typically utilized for IPDA, ~12 km, were employed to increase the DIAL retrieval SNR, a result of weakly scattering atmospheric aerosols and molecules compared to the strong surface signal. The DIAL derived DAOD at the SSE was compared to the standard IPDA and the in-situ derived estimates, showing good agreement with <1% retrieval accuracy. We expand further on these atmospheric retrievals by demonstrating the novel ability to directly apportion the PBL DAOD from the column in clear air conditions using the range resolved DAOD profiles. Comparisons of the HALO derived PBL DAOD/$XCH_4$ to the in-situ derived PBL column showed favorable agreement, on order of 1% absolute difference, and provide a foundation of understanding needed to make $CH_4$ atmospheric profiling an operational product for future campaigns. To enable this, future instrument enhancements include the use of higher sensitivity HgCdTe detectors and further optimized gain settings between the DIAL and IPDA channels. The range resolved DIAL methods presented herein have the potential to provide new insights on $CH_4$ fluxes across scales and offer an avenue for the first remotely sensed profiles of atmospheric $CH_4$ with the needed sensitivity for inventory and survey studies. The added HSRL observations made by HALO also provide unique contextual information that will be critical for validation of future passive $CH_4$ measurements from space.

**Data Availability**

ACT-America observational and modeling datasets are archived at the ORNL DAAC (https://daac.ornl.gov/actamerica) and at https://www-air.larc.nasa.gov/missions/ACT-America/. The ACT-America in-situ aircraft data used in this study can be found at https://doi.org/10.3334/ ORNLDAAC/1556 and https://doi.org/10.3334/ORNLDAAC/1574. HALO $CH_4$ products are not yet available at the ORNL DAAC but are available upon request.

**Author Contribution**

RABG led the analysis presented here with contributions from ARN. SK, JC, and ARN lead data curation and development of the HALO HSRL products. RABG, SAK, ARN, and JEC developed HALO's methane retrievals. RABG, ARN, JEC, DBH, and JL contributed to the preparation and deployment of the HALO instrument. JPD and YC provided in-situ data collected during the campaign for comparison to HALO. KJD led the ACT-America campaign, designed the flights, and directed execution of the flight patterns. RABG prepared the manuscript with contributions from co-authors.

**Competing Interests**

The authors declare that they have no conflict of interest.



**Acknowledgements**

We acknowledge funding support from the NASA Headquarters Earth Science Division, the NASA Earth Science Technology Office, and the NASA Langley Research Center. We thank the C-130 and B200 teams at the NASA Wallops Flight Facility

and NASA Langley Research Center, respectively, and the National Suborbital Education and Research Center for their support of the ACT-America campaign. We acknowledge the use of imagery from the NASA Worldview application (https://worldview.earthdata.nasa.gov/), part of the NASA Earth Observing System Data and Information System (EOSDIS). The Atmospheric Carbon and Transport (ACT)-America project is a NASA Earth Venture Suborbital-2 project funded by NASA's Earth Science Division (Grant NNX15AG76G to Penn State).

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
