# Peer review of "Evaluation of the High Altitude Lidar Observatory Methane Retrievals During the Summer 2019 ACT-America Campaign"

_Atmospheric Measurement Techniques, 2022_

## Author Comment (AC1)

**Response to Anonymous Referee #1 Comment on amt-2022-106 from 05 May 2022**

**The authors would like to thank the reviewer for their helpful comments that have helped improve the quality of this manuscript. The reviewer's comments are listed below in black with our responses given in red.**

**General comments.**

First I would like to mention that this paper show impressive work and results with respect to XCH4 estimate using lidar IPDA and even DIAL methods. The paper is well written and the figures are clear and well detailed which make easy the understanding of the measurements. Although previous XCH4 airborne IPDA measurements with CHARM-F is mentioned some discussion of performances (precision, resolution, biaises) with respect to CHARM-F and maybe future validation of MERLIN space mission is missing.

The suggestion of providing a direct comparison to CHARM-F or MERLIN would be beneficial in the framework of future validation. Within this paper we sought to describe HALO as a new measurement capability within the carbon cycle community with its performance described thoroughly such that comparisons between heritage and emerging measurement techniques/instruments could be carried out by the reader. This paper is intended to form a benchmark of performance for the HALO instrument while operating on an environmentally challenging airborne platform. Direct comparison between other instruments is challenging as the observational conditions are not identical, and in many instances not similar at all. We believe that detailed comparisons to CHARM-F and MERLIN are outside the scope of this paper, however, some references to MERLIN and CHARM-F performance metrics from published papers have been included throughout the manuscript.

The paper is quite long and one may think that the HSRL measurements, as much as a detailed geophysical analysis of XCH4 data should be kept for a second paper.

Although the manuscript covers a lot of topics in detail, we feel the inclusion of the HSRL measurement capability is necessary as it addresses several key issues we wish to portray to the broader community. First, the inclusion of the HSRL measurements demonstrates to the passive remote sensing community that there is now an instrument capable of remotely measuring accurate and precise methane columns for validation while also offering quantitative information on aerosol bias on the passive retrievals. Second, as we note in the paper, the inclusion of the HSRL measurements not only provides context to the XCH4 measurements, but provides qualitative comparison to the aerosol backscatter measurements presented at 1645 nm. We believe the short HSRL introduction section and accompanying panel plots do not add significant length to the manuscript.

As I said the paper seems to me a little long, however, in the same time, a fundamental part is missing: the consideration and correction of statistical biases in DAOD and DIAL estimates. Such omission can lead to misunderstanding of differences between in situ and lidar measurements. Therefore, some of the results in this paper should be calculate again and corrected and this is the reason why I indicated "major revision". I recommend to make the following correction before consideration for publication.

Discussion of statistical biases is now presented within the paper and utilizes many of the references that the reviewer suggested. Further within this response is some supporting analysis that supports our decision that re-analysis of the data used within the paper is not necessary. We believe that the HALO SNRs are sufficiently high such that statistical biases are not confounding our comparisons and conclusions between HALO and in-situ in both the IPDA and DIAL retrievals, which we discuss further below.

**Specific comments.**

- **Instrument:**

- L162. Note that main difference with CHARM-F is the OFF wavelength 1645.86 nm -> 1645.37 nm on the other side of the CH4 multiplet. Recent spectroscopic data on H2O absorption lines [Delahaye JQRST 2019] show that minimising H2O impact on CH4 DAOD requires to use the OFF-line at 1645.86 nm. This should be maybe indicated

or at least taken into account if the authors plan to contribute to a future validation of MERLIN CH4 space lidar mission

We point you to our 2013 simulation paper showing the impact of the offline selection on the XCH4 retrieval: Refaat, Tamer F., et al. "Performance evaluation of a 1.6-µm methane DIAL system from ground, aircraft and UAV platforms." *Optics express* 21.25 (2013): 30415-30432.

The off-line wavelength was selected on the shorter wavelength side of the line center. This new off-line location provides less sensitivity to $H_2O$ and $CO_2$ and reduces the online-to-offline separation (179.4 pm). A reduced online-to-offline separation can help to minimize systematic errors due to differential aerosol scattering and extinction for range resolved DIAL and differential albedo effects for IPDA. Figure 3 from Refaat et al. 2013 shows the advantage of the selected off-line location in minimizing $H_2O$ differential optical depth near the surface, where the gas amounts are largest and most variable. This minimizes the interference from $H_2O$ variability on the CH4 measurements. This previous analysis provides the motivation for operating the HALO CH4 offline at the current location of 1645.3724 nm, where the differential OD between the two-line locations is minimized. The following text was updated in accordance:

"The reduced impact from $\delta\tau_{H_2O}$ results from optimal offline wavelength selection such that $\delta\tau_{H_2O}$ is minimized near the surface (Refaat et al. 2013)."

As the reviewer indicates, future MERLIN validation efforts will benefit from minimizing differences in spectroscopy. This can easily be accommodated by locking the HALO offline wavelength to the same spectral location as MERLIN/CHARM-F. In fact, operating on both sides of the line within the scope of a future validation effort will provide useful information for the optimization of future active sensing missions.

- **Opical depth biais correction:**

- L334 and Figure 6c. Why the biais correction depends on the gain Figure 6c ? I don't see a potential explanation in all is described L334-342.

Each gain channel has different differential transmission and detection chain non-linearities that can be imparted onto the signal and thus the DAOD retrieval. Low SNR present on the LOLE channel (low gain) also contributes to the differences seen in Fig. 6c. The following text was added:

"In addition to differences in SNR and uncertainty in spectroscopy, fractional differences between HALO and in-situ truth can result from differences in the differential transmission between the different optical channels and different differential transient responses between the different electrical gain channels."

Moreover, as the main difference of biais correction is shown for the LOLE channel (with the lowest SNR) one may wonder if the necessary correction of averaged optical depth with low SNR have been taken into account in the signal processing (Tellier et al. AMT, 2018) ?

It is important to note that although the correction shown in Fig. 6c includes data from all altitudes and for all gain channels (i.e. a wide range of SNRs is encompassed), the data employed for the retrievals presented in this paper are selected from a gain channel only when sufficient SNR is achieved and the next highest gain channel was non-linear or saturated (i.e. unusable for linear retrievals). Calibration of the LOLE channel, where high SNR is experienced, while operating in the 'unattenuated' modality was carried out later in the campaign using the methods shown in Fig. 6. We show below that for IPDA retrievals, regardless of which channel is used, SNR is sufficiently large such that statistical bias is negligible.
Additional text was added within the paper around Fig. 6 to clarify the methods used to calibrate the different gain channels and emphasize the use of signals when the SNR is appropriate.

The reviewer notes that nothing is said about statistical biaises in IPDA/DIAL in theoretical paragraph 2.2 and further in the paper which is not acceptable.

We recognize the importance of statistical biases and have included the topic of statistical biases in IPDA/DIAL. The following text was provided within the altitude dependent bias discussion in Section 3.1.1:

"Recent studies have indicated that statistical and geophysical biases can also manifest from low SNR retrievals or from sufficiently long along track averaging, though corrections have been developed for each (Tellier et al. 2018). Initial assessment of HALO's native 0.5 s retrievals for each gain channel, found that optimized receivers gain exhibiting high SNR displayed negligible $CH_4$ DAOD bias, <1e-3 in DAOD."

**If no statistical biais is considered, the a posteriori biais correction used by the authors in paragraph 3.1.1 is clearly SNR dependent** and this should be indicated even if it is not obvious in the DAOD estimates.

As discussed above, we employ different gain channels only when sufficient SNR is achieved, and the next highest gain channel is saturated or suffers from non-linearity. Although the LOLE signal is presented in Figure 6 for all altitudes (reminder that the signal is attenuated as the aircraft descends), it is only employed for low altitudes and over high albedo targets where the signal from other channels is saturated and SNR is sufficiently high to yield an acceptable retrieval. To provide insight on the impact of SNR shown in Fig. 6 we calculate the statistical bias per optical depth measurement as a function of the online and offline SNRs according to Tellier et al. 2018, as $bias(\delta\tau_{CH_4}) = \frac{1}{4}\left[\frac{1}{SNR_{on}^2} - \frac{1}{SNR_{off}^2}\right]$. We do not collect single shot data due to file size constraints, so the highest temporal cadence that this metric can be calculated is 2 Hz, or 250 online/offline summed shot pairs. The single shot SNR can be estimated from the 250 shot sums by dividing the 250 shot SNR by $\sqrt{250}$. Figure 1, below, shows the SNR and resulting bias calculated for all gain channels from Fig. 6 of the paper at the native 2 Hz rate (250 accumulated shots). The distributions are for all four altitude legs combined. For the optimized SNR high gain channel (single shot SNRs on order of 100 and 2Hz SNR of ~1500), the bias is negligible, however for the non-optimized low gain channel the bias is higher. In this scenario the high gain channel is used for the $XCH_4$ retrievals. The low gain LOLE signals are only employed for $XCH_4$ retrievals when they are optimized at lower flight altitudes or where the range to the scattering target is sufficiently small. This ensures sufficiently high SNR to yield low bias and high precision. The calculations in Fig. 1 provide an indication that the SNR is sufficiently high for the optimized gain channel and the induced statistical bias is negligible. Therefore, given the high SNR retrievals for these calibration maneuvers, it is acceptable to assume that most of the disagreement between the lidar and in-situ truth stems from uncertainty in the spectroscopic parameters used for the retrieval and not directly from statistical bias.

[Figure]

*Figure 1 – SNR and statistical bias on the IPDA retrieval for 250 online(blue)/offline(orange) shot pairs for the Fig. 6 stepped altitude calibration maneuver. The mean SNR for each wavelength is greater than 500 for the optimized high gain, with a resulting calculated bias of <<1%. This indicates that the statistical bias is negligible for the high gain optimized retrieval. From panel (a), the optimized gain for this maneuver, the distribution mean of the online is an SNR~1225 and the mean of the offline is an SNR~1675 for 250 shots, for single shot this gives an online mean of SNR~77 and an offline mean of SNR~105.*

- L406. Precision performances of HALO should be compared and discussed with respect to Amediek et al. AO 2017 paper and CHAM-F results.

      As discussed above, we do not feel it is necessary to provide a direct comparison to another instrument as this is a paper benchmarking the capabilities of a new instrument for application to future airborne investigations, however, the authors have added the following additional text to the paper to reflect a comparison to CHARM-F:

      "For HALO retrievals utilizing a gain channel that results in an optimal SNR, the HALO DAOD and resultant $XCH_4$ retrievals show comparable results for similar averaging scales to those previously published on CHARM-F (Amediek et al. 2017)."

- Figure 9 and L422. XCH4 noise statistic decreases less than the square root law. We can find a similar result in Amediek et al. 2017. The reason that is suggested by the authors is « harsh operating condition in the C-130 » and « high vibrational environment » which is fully possible and may entail a « degraded laser frequency stability » … what about optical misalignment issue? did the authors make some vibrational tests of their system?

      We appreciate the comment on Amediek et al. 2017. We do not believe that optical misalignment is the root cause as engineering flights on other aircraft (with noise statistics shown in Nehrir et al. 2018) did not exhibit the same degree of impact. Additionally, during the development of the HALO laser transmitter significant vibrational testing was performed and has been presented with Fitzpatrick et al. 2019. Additionally, HALO employs an active boresight channel which ensures that the transmitted laser is maintained within the center of the telescope field of regard to minimize any albedo or vignetting effects that could arise from system level misalignments. Additionally, pupil imaging is employed in the receiver to ensure that any beam misalignment (albeit very minor and on order of 10s of microradians) during the automatic boresighting does not result in changes in the spatial distribution of received light on the detector. This approach minimizes impacts resulting from non-uniformity of gain across the active area of the detector. This leads us to believe that the impact to the averaging statistics is a result of the platforms effect on the instrument. We have now added text which includes this reference to prior vibrational testing and results. See the following modified text:

      "Although high precision can be achieved with relatively short averaging times, and different gains are employed to allow operational flexibility, the performance observed during ACT-America fell short of prior flights on the Langley B200 aircraft (Nehrir et al. 2018). The increased statistical noise observed could result from the harsh operating conditions on the C-130, resulting from slightly degraded laser frequency stability due to the high vibration environment. Dedicated structural thermal and optical analysis of the laser transmitter subsystem was performed prior to full instrument test flights but did not indicate a significant degradation of performance (Fitzpatrick et al. 2019)."

- **Regional scale observations**

- Figure 13. and L530-535. The unexpected result of larger IPDA XCH4 than PBL in situ CH4 is very unusual. This shows that in situ data measured both in the free troposphere and in the PBL may not be sufficient to make a validation of space-based measurement such as MERLIN. Co-located airborne measurement and maybe XCH4 profiling with ground-based lidar should be used to explain such enhancement of CH4 in the free troposphere.

      We agree that the enhancement of HALO over the in-situ measurements is unusual and suggests free-troposphere enhancements. Such enhancements have been observed in a wide range of airborne field campaigns. We agree that the in-situ observations at point locations in different parts of the atmosphere do not provide enough information for validation of columnar observations such as those from HALO, CHARM-F, MERLIN, and other passive spectrometers. A true validation effort for any airborne or space-based column integrating instrument (active or passive) will require a dedicated aircraft for carrying out in-situ profiles. Although the bulk of the in-situ observations from ACT-America were collected at fixed altitudes, many coordinated profiles with the C-130 lidar overpass were also collected which provided the unique opportunity for validation that is presented herein. One important note for future validation efforts is to ensure that the in-situ aircraft samples the lowest part of the atmosphere either through a missed approach at an airport or through other means (point measurements at the surface).

- I think that the second part L550 to L600 is not necessary in this paper (although really interesting!). Also, HSRL measurements seem not to be so essential in this paper as the authors proved that 1.645 μm backscatter is sufficient to give the vertical structure of the atmosphere and even, I guess, the height of the PBL.

We respectfully disagree and feel that this section of the paper provides much needed context to the first spatial comparison between HALO and in-situ observations. The spatial correlation figures between the remote and in-situ data demonstrate the sensitivity of the columnar observation from the lidar to small variability within the PBL where fluxes dominate the signal. We believe that showing this comparison is important to highlight biogenic signals, contrasting the anthropogenic signals in the prior comparison. Additionally, the HSRL observations presented here as well as the supporting text offer to the reader and the broader community potentials for how these observations could be leveraged for optimization of future airborne missions for surveys, process studies and cal/val activities.

**Advanced CH4 products - Atmospheric profiling**

L 646- 663. and Figure 17

No change was made as this line is referring specifically to Fig. 16c range resolved DAOD curtain, where the vertical resolution is 350 m and the along track resolution is at 0.5s (not yet averaged in time along track and shown in Fig. 17b).

- L 646. The estimate of the DAOD profile is confusing. Did the author slice average the backscattered profiles to 350 m and 15 s first before using equation 6 ? It does not look this way given the variations of DAOD profile in Figure 17b… Signal processing should be clarified here.

The profiles of backscatter are combined first in time along track and then in range vertically to give single profiles for the online and offline backscatter at 15 s horizontal and 350 m vertical resolutions. These averaged profiles are then used within equation 6. This text was modified in the paper to:

"The online and offline backscatter profiles were first averaged 15 s along track (2 km) and then to 350 m in the vertical prior to use in Eq. 6., shown in Fig. 17a"

- A decreasing DAOD is of course not expected and I agree that this may be a manifestation of low SNR. I have then the same question as for IPDA measurement: did the authors make an estimate of the statistical biais on the DAOD (and thus a correction) due to the non-linearity of equation 6?  this question is linked to the question just above giving that an averaging enables to increases the SNR and then to decrease such biais… once again please read Tellier et al. AMT 2018 but this issue was also mentioned in early DIAL measurements with high precision such as for CO2 (Gibert et al. JTECH 2008)

We overlooked the range resolved $CO_2$ optical depths measurements made in Gibert et al 2006 and Gibert et al 2008, thank you for bringing them to our attention. We have added these as references to the manuscript:

- Gibert, Fabien, et al. "Two-micrometer heterodyne differential absorption lidar measurements of the atmospheric CO 2 mixing ratio in the boundary layer." *Applied optics* 45.18 (2006): 4448-4458.
- Gibert, Fabien, et al. "Vertical 2-μm heterodyne differential absorption lidar measurements of mean CO2 mixing ratio in the troposphere." *Journal of Atmospheric and Oceanic Technology* 25.9 (2008): 1477-1497.

Examining the averaged data used within the computation of Figure 17 b, the 12.5 km along track by 350 m vertical resolution results in high SNR in the PBL where we make the clear air retrieval. The SNR results are shown below this response in Fig. 2. The DAOD bias is low, in all regions of the profile, specifically within the boundary layer where the lowest atmospheric retrieval bin resides. Because we are utilizing accumulated optical depth within the boundary layer where SNR is high, the noisy or low SNR DAOD in the middle of the profile does not impact the clear air PBL retrievals as indicated and therefore the impact of statistical bias on the clear air PBL DAOD is negligible.

[Figure]

*Figure 2 – SNR of the 350m vertical by 12.5 km backscatter profiles and calculated statistical bias for the range resolved DAOD calculation from Fig. 17 of the paper.*

As for IPDA, a correction of DAOD with statistical biais is a basis for modern DIAL measurements and the authors should includes and discuss in details the impact on SNR on their measurements. This is to my mind essential.

However the authors should be aware that the correction of DAOD with SNR linked biais might not be sufficient to remove entirely the decrease of DAOD seen in Figure 17b. At low SNR, especially for ON line signal the impact of Pb removal in Equation 1 may entail other issue due to the electronic baseline and linearity of the detection.

Text discussing the impact of SNR on the range resolved profiles of DAOD has been added to this section of paper. We do not believe that a correction of DAOD with statistical bias is necessary, as the SNR of both the DIAL (in the region of interest) and IPDA retrievals are well above the regime in which those systematic effects start to become important. The level at which DAOD statistical bias becomes a concern is on the low end where SNR is <10. We show above that the SNR for DIAL (Fig. 2) and IPDA retrievals (Fig. 3) are sufficiently high such that the impact of statistical bias in negligible.

The following text has been added to address statistical biases:

"The impact of low SNR can manifest as a statistical induced DAOD bias within the associated retrieval bin when utilizing Eq. 6. This effect has been well documented within Gibert et al. (Gibert et al. 2006,2008), which indicate that the magnitude of the bias can be considered negligible for high SNR backscatter (SNR>10) that have aggregated signal over multiple shots and or multiple range bins (Gibert et al. 2006). It was found that the averaged profiles of backscatter exhibit high SNR throughout the majority of the profile. Examining Fig. 17b, the online wavelength's SNR at the top of the PBL is ~1000 and the SNR at the low backscatter feature at ~2.4 km is ~80. This gives an indication that the regions of interest near the surface and within the PBL exhibit higher precision due to the higher per bin SNR for each wavelength."

- L 658. A linear regression on the DAOD that is not weighted by error bars on each DAOD point is biased for the reason mentioned above and non linearity of equation 6. Gibert et al. AO 2006 used such likelihood estimate to make accurate XCO2 measurement in the PBL. In Figure 17b the DAOD will then not impact so much a likelihood calculated slope coefficient and I expect that there will be a better agreement with IPDA and in situ DAOD.

In conclusion the difference is not at this point explained by spectroscopy as the authors wrote (this sentence should be removed) but clearly by the non consideration of statistical biaises in their estimates.

The authors have examined the Gibert et al. 2006 results on range profiling CO2 DAOD, specifically Figures 7 and 8. Of interest is the finding in Fig. 8 which emphasizes the reduction of statistical uncertainty as a result of shot averaging. The range resolved DAOD calculation performed in Fig. 17b here is identical to that within Gibert et al.

2006, and the input online/offline shot pairs for the 12.5 km along track resolution surpass the point at which an impact would be significant. Combined with the analysis shown above, we are confident that the fitting routine and extrapolation to the surface is not biased by low SNR to pose a prominent impact on the DIAL DAOD ground estimate. We believe in this case, spectroscopy (either as transmit frequency error or spectral impurity) is likely the cause for overestimate. Additionally, results from Gilbert et al. 2006 are not directly comparable to those presented here for numerous reasons. The observing geometry is reversed compared to what we present here, which is important as for DIAL and IPDA the SNR is proportional to the DAOD. This specific point is important as the highest SNR (CNR) presented in Gilbert et al. 2006 is associated with the smallest DAOD, such that as DAOD is accumulated with increasing range from the lidar the SNR is significantly reduced. In this geometry/measurement configuration, the reviewer is correct, the regression requires weighting of the higher SNR observations closer to the instrument (lower DAOD) over those further in range where DAOD is largest and uncertainty is as well. For HALO the opposite is true, where our lowest DAOD is balanced by high SNR near the aircraft and the highest DAOD near the surface is complimented by enhanced SNR due to higher backscatter within the PBL. Therefore, the linear fit is bookended by high SNR observations and the impact of lower SNR measurements in the middle of the profile (which still exhibit significantly higher SNR than those presented in Gilbert et al. 2006) do not negatively impact the extrapolation to the surface.

- L710 - 720. Of course what is mentioned above should be considered in all this paragraph, i.e. the different statistical biaises should be corrected before the comparison of PBL XCH4 using the cloud slicing method and the DIAL profiles.

As already said before, spectroscopy induced error should be mentioned, if necessary, only in a second step.

For the cloud slicing analysis shown in Fig. 18 a,b of the paper, we have performed the SNR analysis and respective bias calculations for the highest native resolution (half second averaged – 250 shot pairs per retrieval) returns, shown below in Fig. 3. The IPDA results here employ the LOLE channel in the unattenuated modality. Unfortunately, operating in unattenuated mode in an attempt to maximize opportunity for range-resolved profiling rendered the medium and high gain channels (HOLE and LOHE) unusable due to saturation (we have since reoptimized the gain settings on the different channels to allow us to better utilize the higher gain channels over a larger dynamic range). For this specific comment, we see that for both the cloud and ground returns on the LOLE channel that the respective SNR per wavelength is high and the resulting bias is low, and would significantly decrease with additional averaging as employed throughout the paper (Fig. 15 a,b shows 15 second averages). Given this analysis we elected to not perform any additional bias correction as suggested to the IPDA retrievals for the PBL XCH4 estimates. For additional context, Figure 4 below shows the SNR for the LOLE channel during one of the spiral overpasses shown in Figure 11 of the paper. Here we demonstrate again that the LOLE channel has commendable SNR at the native 2Hz sampling resolution to the point where statistical bias can be neglected.

[Figure]

*Figure 3 – Top panels(a,b): Calculated SNR of the cloud and ground returns for the IPDA returns used within the cloud slicing retrievals shown in Fig. 18a,b of the paper. High SNR can be seen in the cloud and ground returns. Bottom panels (c,d): the associated bias calculated for each of the cloud and ground returns. Minimal impact can be seen relative to the respective DAOD.*

[Figure]

*Figure 4 – Top panel (a): C-130 flight altitude and DEM altitude from a spiral overpass from July 26,2019.  Bottom Panels (b,c): Calculated SNR and associated bias for the IPDA signals from the LOLE channel from a spiral overpass on July 26 2019.*

**Technical corrections**

- L390. please remove one « is » in the sentence.

     Fixed.

- L696. Please add an error bar for each retrieval IPDA ground, cloud, PBL

     We have added error bars to the DIAL retrieval in Fig. 18a. The histograms in Fig 18b provide an indication of the random error in the IPDA retrieval for ground and cloud top. Additional 1-sigma error bars in 18a for the IPDA measurement, beyond the histograms, will clutter the figure and make the figure hard to interpret.

- 5.1 paragraph. As there is no 5.2 paragraph I guess that this title should be removed

     We chose to keep this to indicate a subsection of the greater section 5.

- Conclusion must be re-written in agreement with statistical bias corrected results.

     We have changed the conclusion to reflect the additional discussion spurred from this review.

---

## Author Comment (AC2)

**Response to Anonymous Referee #2 Comment on amt-2022-106 from 05 May 2022**

**The authors would like to thank the reviewer for their helpful comments that have helped improve the quality of this manuscript. The reviewer's comments are listed below in black with our responses given in red.**

This paper describes the first results from a lidar system deployed onboard a research aircraft measuring atmospheric methane. The paper is well-written, and fits well within the scope of AMT. However, a few issues listed below should be addressed before the paper can be recommended for publication.

General comments:
Note that the term "mole fraction" is recommended rather than "mixing ratio", see e.g. https://www.empa.ch/web/s503/gaw_glossary#recommendations. I suggest simply replacing throughout the text. Also I would recommend to consistently use ppb for dry air mole fractions of CH4. Using both ppb and ppm (e.g. Fig. 10 (b) ) is confusing to the reader.

We recognize this difference and appreciate the reviewer's input. We have changed references of "mixing ratio" to "mole fraction" throughout the paper. We adjusted the units in Fig. 10b to be completely in PPM.

Comparison to in-situ measurements: The deployment of the different aircraft sampling different altitude regimes really has potential, as indicated e.g. by Figs. 12 and 13 and the associated discussion. I suggest a simple combination of the in-situ measurements within the free troposphere from the C-130 aircraft, the boundary layer in-situ measurements from the B200 aircraft, and the estimate of the boundary layer height derived from the HSRL measurements onboard the C-130, to calculate a partial column XCH4 based on insitu observations that can directly be compared to HALO XCH4. The assumption is that CH4 is well mixed within the PBL and also within the free troposphere. Any advection of air masses with enhanced CH4 above the PBL would clearly stick out as differences between HALO XCH4 and the aircraft derived XCH4.

We recognize how the suggested analysis would be beneficial to understand the partial free troposphere vs boundary layer columns, however for the along track segments shown in Figs. 12 & 13 there are no in-situ spirals that can be leveraged to apportion the different parts of the column as suggested. In the future, we hope to have the necessary SNR from the lidar data to provide a robust assessment of the various partial columns for specific regions such as these. We will indicate, however, that we do show an in-situ spiral column in Figure 10, where the in-situ derived XCH4 is calculated from various altitudes, so the total column and partial columns can be seen. Additionally, the analysis shown in Figure 18 demonstrates apportionment of the PBL and provides an indication of HALO vs. in-situ.

Dry air mole fraction - impact from H2O: In the in-situ measurement community there is much discussion on drying/conditioning samples before measurement vs. correcting based on simultaneous H2O measurement within the exact same sample. As the authors describe, MERRA humidity is used in the retrieval of XCH4 (the column average dry air mole fraction). The uncertainty in XCH4 introduced by this choice should be assessed, e.g. by comparing MERRA water vapor to that of the in-situ observations.

The topic of XCH4 impact from H2O presence at the online/offline transmitted wavelengths was also brought up in Reviewer Comment #1. Due to the selection of our offline wavelength, the differential optical depth due to water vapor is minimized near the surface where most of the water vapor and water vapor variability resides within the troposphere. This method of minimizing the differential absorption as opposed to minimizing the differential cross section makes the impact of water vapor on the XCH4 retrievals negligible. This implies that regardless of the use of MERRA or in-situ profiles of water vapor, the latter of which are only available during spiral ascent/descents, the total impact of using MERRA versus an in-situ profile is small on the total propagated impact on the XCH4 retrieval.

[Figure]

*Figure 1 - Comparison of the in-situ spiral generated water vapor profile to the mean profile of water vapor number density generated with MERRA-2 reanalysis for the duration of the spiral overpass.*

To support this conclusion, we have included some additional analysis here derived from the July 20th spiral profile shown in Fig. 10b of the manuscript. In Fig. 1 above we have taken the in-situ profile of water vapor number density generated during the spiral and compared to the mean MERRA reanalysis for the spiral overpass. Using these profiles to calculate the water vapor DAOD (pressure and temperature were utilized from each respective source for the computations) we calculate that the DAOD for in-situ is approximately 2.61e-4 and MERRA is 3.00e-4, both falling within 13% of each other, but ~0.09 % of the $CH_4$ DAOD. Utilizing Eq. 3 from the manuscript, the difference in water vapor DAOD to the $XCH_4$ retrieval between MERRA and in-situ yields an approximate 0.2 ppb difference. We feel that this is well within the random and systematic uncertainty bounds of our measurement capability and lends credence to the validity in utilizing the MERRA reanalysis for relative humidity inputs. It is necessary to note that under standard flight operations, void of spiral generated in-situ profiles, we report $XCH4$ using reanalysis atmospheric state to encompass all sources of error, this includes utilizing the relative humidity product.

The following text was added to Section 2.2 within the discussion of MERRA-2 atmospheric state:

"Comparisons of retrievals using MERRA-2 atmospheric state to those using in-situ profiles from spiral maneuvers indicate that differences are <1 ppb. Use of MERRA-2 under normal flight operations serves to then include atmospheric state error within the XCH4 retrieval, as expected for retrievals made in all regions without access to in-situ profiles."

Specific comments:
L271 "over samples" -> "oversamples"

Fixed

L280 "Altitude is used in lieu of MSL for all figures" this is not clear. May be "Altitude above MSL is used in lieu of Altitude above ground level"?

We have adjusted this for clarity.

L280: "post- flight reanalysis" may be drop "post-flight"? I guess reanalysis products are available only for past periods, i.e. after the flights, anyway.

The authors agree and have removed post-flight.

L331: "The superscript will be dropped for simplicity." Which superscript?

We have adjusted this for clarity: "The 'cal' superscript will be dropped for simplicity."

L362: the matrix T should contain the elements that the beta-vector elements are multiplier with, i.e. 0th, 1st 2nd and 3rdorder terms as formulated in the Eq. on line 361. May be simply write down the first and last row of the matrix, and the few elements

We recognize that this comment is the result of not enough information describing the matrix $T$. We have taken the liberty to provide more description as follows:

" $T = \begin{bmatrix} 1 & \delta\bar{\tau}_{CH_4} & \delta\bar{\tau}^2_{CH_4} & \delta\bar{\tau}^3_{CH_4} \\ \vdots & \vdots & \vdots & \vdots \\ 1 & \delta\bar{\tau}_{CH_4} & \delta\bar{\tau}^2_{CH_4} & \delta\bar{\tau}^3_{CH_4} \end{bmatrix}$ is the matrix composed of $\delta\bar{\tau}_{CH_4}$ "

Fig. 6: please adjust color selection for the different gains so that color blind people can read the figure. To me HOLE and LOLE are identical, LOLE is very slightly different.

We appreciate the input here. We have changed the color diagrams for Figure 6 utilizing: https://yoshke.org/blog/essays/2020/07/colorblind-friendly-diagrams/ to accommodate color blind people (RGB [204,121,167];[0,114,178];[86,180,233]).

Fig. 6 caption: please explain DEM (I know what it is, but it should be mentioned once)

We have adjusted this for clarity, though it is also mentioned within the text.

Fig. 11: the Y-intercept is not clear. It should be negative, given the slope is larger than one, and the regression line crosses the 1:1 line at around 1900 ppb.

The reviewer is correct, since this the slope is greater than 1 the y-intercept from the fitting routine is negative. The fit provides y=1.0922x – 0.1755. In the scope of the XCH$_4$ measurements presented this y-intercept does not have a tangible meaning, so we had previously attributed the intercept to where the regression line crossed the plot axes. We have updated Fig. 11 to show the fit parameters directly.

L522: "PA region" – to make this clear to non-US readers (AMT it is a European journal) may be add a label to Figs 11 (a) and (c)

We have adjusted this for clarity in the text and provided an indication in Figure 12a,c.

L571: I don't see any cross-hatched area in Fig. 14, may be I am misunderstanding something

We have altered the wording here to be more descriptive of the region we wish to emphasize.

"Here all instruments register enhancements, emphasized within the lower flight track section of Fig. 14b."

L612 "spatial" -> "spatially" or drop

"spatial" was dropped due to redundancy.

L659: the dial DAOD estimated at SSE shown in the inset of Fig. 17 (b) (magenta symbol) is around 0.2875, not at 0.9243 as given in the text. Please clarify.

This was an error on our part, we have gone ahead and replaced the figure with the correctly indicated DAOD retrieval. We appreciate the reviewer catching this.

L661: "un-bias-corrected" may be use non-bias-corrected

Modified to read "non-bias-corrected", which we agree is grammatically correct.

L736 "PBL fluxes" use PBL mole fractions or concentrations

We changed this to mole fractions for correct description.